# LLMs – the Good, the Bad or the Indispensable?: A Use Case on Legal Statute Prediction and Legal Judgment Prediction on Indian Court Cases

**Shaurya Vats**
IIT Kharagpur

**Atharva Zope**
IISER Kolkata

**Somsubhra De**
IIT Madras

**Anurag Sharma**
IISER Kolkata

**Upal Bhattacharya**
IISER Kolkata

**Shubham Kumar Nigam**
IIT Kanpur

**Shouvik Kumar Guha**
NUJS

**Koustav Rudra**
IIT Kharagpur

**Kripabandhu Ghosh**
IISER Kolkata

## Abstract

The Large Language Models (LLMs) have impacted many real-life tasks. To examine the efficacy of LLMs in a high-stake domain like law, we have applied state-of-the-art LLMs for two popular tasks: Statute Prediction and Judgment Prediction, on Indian Supreme Court cases. We see that while LLMs exhibit excellent predictive performance in Statute Prediction, their performance dips in Judgment Prediction when compared with many standard models. The explanations generated by LLMs (along with prediction) are of moderate to decent quality. We also see evidence of gender and religious bias in the LLM-predicted results. In addition, we present a note from a senior legal expert on the ethical concerns of deploying LLMs in these critical legal tasks.

## 1 Introduction

The Large Language Models (in short, the LLMs) have touched upon many AI-based tasks. There has been a recent surge in the application of LLMs e.g. GPT (Brown et al., 2020), BLOOM (blo), FLAN-T5(t5), LLaMA(lla) which has heralded a new area in data science. Along with many other areas, LLMs have been applied to the legal domain. Blair-Stanek et al. (2023) has applied GPT-3 for statutory reasoning on U.S. statutes with moderate success. Similarly, (Yu et al., 2022)(Nguyen et al., 2023) applied GPT-3 for the legal reasoning task of statute detection on a Japanese Bar exam dataset effectively. Katz et al. (2023) shows that GPT-4 is able to pass the Uniform Bar Examination (UBE), often outperforming humans. However, (Choi et al., 2023) reported that ChatGPT performs moderately on real exams at the University of Minnesota Law School. (Hamilton, 2023) has designed a multi-agent system for judgment prediction using GPT-2 for U.S. Supreme Court with moderate accuracy. As a different line of work, Savelka (2023) leveraged GPT for sentence-level annotation of legal

text. Recently, (Charlotin, 2023) presented a detailed take on the future of law in the light of the large Language Models.

However, the acceptability of LLMs in a high-stake domain like law is still under scrutiny. To this end, we address the following in this paper: (i) We attempt two legal-specific tasks – statute prediction (Section 2) and judgment prediction (Section 3), using LLMs in the light of state-of-the-art baselines, on Indian Supreme Court cases. (ii) We generate explanations for each of these tasks using LLMs and compare them with *annotated explanations* by legal experts, providing detailed quantitative and qualitative analysis of the same (Sections 2 and 3). We also identify errors and hallucinations in the generated explanations (Tables 11 and 12). (iii) We also discuss the presence of bias in the outputs generated by LLMs and ethical considerations for LLMs in law (Section 4). (iv) We also release a legal-expert-explanation-annotated statute prediction dataset. To our knowledge, this is among the first works so far on Indian cases. In a recent work, (Deroy et al., 2023) applied LLMs to legal court case summarization. Also, to our knowledge, no prior work has addressed these issues on these legal problems esp. for Indian cases. The data and code are available here: https://github.com/somsubhra04/LLM_Legal_Prompt_Generation.

## 2 Statute Prediction

Given a fact text (Bhattacharya et al., 2019b) of a case, we consider two types of tasks (i) prediction of statutes (statute predn) and (ii) prediction of statutes along with the generation of explanation for the prediction (statute predn + expln).

### 2.1 Datasets

**statute predn + expln test dataset**: We have annotated the FIRE AILA 2019 (Bhattacharya et al., 2019a) Task 2 (Statute Retrieval task) dataset with explanations (see example in Table 4). This dataset

has 45 fact texts from Indian Supreme Court cases and the explanation annotations are done by law students of a reputed law university in India. We plan to release this dataset upon the acceptance of the paper.

`statute predn test dataset`: We randomly select 100 cases (excluding `statute predn + expln` test dataset cases) from the Indian Supreme Court collection. In addition, we select 100 more cases from Indian Supreme Court (excluding already considered 145 cases) containing gender-specific statutes and religion-specific statutes from the Indian law (See Table 13 for a list of these statutes curated after consultation with legal experts) for analysis of religion and gender bias in the statute prediction task (Section 4). These 245 (45+100+100) cases form the test dataset for `statute predn`.

`statute predn train dataset`: This is created by randomly selecting 18021 cases excluding the `statute predn` test dataset cases from Indian Supreme Court collection.

## 2.2 Experimental Setup and Results

**LLM models used:** We applied several models like GPT-3.5 turbo, text-davinci-003, text-davinci-002, davinci (ope), flan-t5-large (t5), bloom-560m (blo), gpt2 (gpt) etc. We observed that the best results on our dataset are provided by text-davinci-003 (paid version) on a random set of 5 cases (not a part of the train-test setup) and so all results reported in this paper are produced by this model.

**Prompts used:** We use Template 2 (Table 6 in appendix) to which, in *zero-shot* setup, texts of applicable statutes with the ids were shown (which does not use `statute predn train dataset`) and for test, given fact texts, the LLMs were asked to generate the applicable statutes (from the already shown ones). For explanation generation, additionally, an appropriate request was done. In addition, we have used Chain of Thought Prompting (Wei et al., 2022) as described in Appendix F. We have also reported additional experiments in the Appendix H.

**Baselines:** We deployed InLegalBERT (Paul et al., 2023), XLNet (Yang et al., 2019) and LEGAL-BERT (Chalkidis et al., 2020) which were fine-tuned for 30 epochs with a learning rate of $3e - 5$ using Adam (Kingma and Ba, 2014) optimization on `statute predn train dataset`. It is to be noted, that it is a multi-label classification problem on 80 statutes.

**Prediction results:** The results of these experiments are reported in Table 1. We report Macro Precision, Macro Recall and Macro F1 on `statute predn` test dataset and `statute predn + expln` test dataset. Note that, **Template 2 prompt** is was applied on `statute predn + expln` test dataset (without the explanation part) for prediction (shown as "Prediction only (45 cases)") and also while generating explanations for the prediction (shown as "Prediction with explanation (45 cases)"). We observe that when the LLM model is prompted to generate explanations along with the prediction, its overall performance (Macro F1) increases. We see that LLM outperforms all the baselines in the prediction task and it is *statistically significantly better* ($p$-value $< 0.05$) than the best-performing baseline (XLNet) by two-sided t-test (tte) at 95% confidence level. Template 2 in Table 6 provides the structure of the prompts given to the LLM for statute prediction which provides, the titles and descriptions of the relevant statutes. This may have been an advantage of the LLM model over the baselines which does not consider the text of statutes. More baseline results are reported in Appendix G.

**Explanation results:** We have also evaluated the explanations generated by LLM by comparing them with gold standard explanations annotated by legal experts. To this end, we use several evaluation measures. We use the n-gram or word-overlap based measures like ROUGE-1, ROUGE-2, ROUGE-L (Lin, 2004), METEOR (Banerjee and Lavie, 2005) and BLEU (Papineni et al., 2002). In addition, we use BERTScore (Zhang et al., 2020) that captures the semantic similarity between the gold standard and LLM-generated explanations. We have also used a recent measure, viz. BLANC (Vasilyev et al., 2020) to estimate the quality of the generated explanations without the gold standard. The results are shown in Table 2. We see that the BERTScore values are better which possibly indicates good semantic match not captured by verbatim match. Table 7 shows some decent explanations generated by LLM in comparison with the expert-labeled annotations.

**Expert opinion on LLM-generated explanations:** We randomly selected 13 cases from `statute predn` test dataset (which do not have annotated explanations) and sent the explanations generated by LLMs on these to a senior legal expert for evaluation. The expert was asked to comment on the

quality of explanations and rate them on a Likert scale of 0-10 (0 – completely unsatisfactory, 10 – perfect). According to the expert, these explanations are of moderate quality with an average score of 4.27 out of 10.

**Hallucinations:** Tables 11 and 12 show some common hallucinations found in LLM predictions and generated explanations. The LLM tends to incorporate statutes that appear more frequently in the contextual examples, even if they are not relevant or applicable to the given context and attempts to provide explanations for the fact statement by utilizing words from the description of the statute provided in the prompt. Furthermore, it prioritizes including the statutes applicable to the fact statement provided in the prompt rather than those relevant to the test case. In order to utilize all the available tokens in the completion, the model aims to include as many statutes as possible, resulting in a higher number of statutes being included in its response. The performance of the model tends to decline when presented with a large test input. As the model progresses down the list of statutes provided in the prompt, its performance tends to decline. This suggests that the model's ability to generate accurate and relevant responses is affected by the order in which the statutes are presented, with the later statutes being less effectively utilized in generating appropriate outputs.

| Evaluation measure | Statute Prediction (45 cases) | Judgement Prediction (54 cases) |
|---|---|---|
| ROUGE-1 | 0.2522 | 0.5490 |
| ROUGE-2 | 0.1934 | 0.4217 |
| ROUGE-L | 0.2445 | 0.5268 |
| METEOR | 0.2362 | 0.4685 |
| BLEU | 0.1043 | 0.2745 |
| BERTScore | 0.3986 | 0.8970 |
| BLANC | 0.2024 | 0.3394 |

Table 2: Explanation performance (F1-score) for Statute Prediction and Judgement Prediction.

ation of explanation for the prediction (judgment predn + expln). We follow the setup of Malik et al. (2021).

### 3.1 Datasets

**judgment predn + expln test dataset**: We have used the 54 (out of 56) annotated cases (ILDC_expert) of Malik et al. (2021) as the remaining two cases will be used for in-context training of the LLM.

**judgment predn test dataset**: We randomly select 100 cases (excluding judgment predn + expln test dataset) from ILDC_single of Malik et al. (2021). In addition, we select 100 more cases ILDC_single of Malik et al. (2021) (excluding already considered 100 cases and 56 cases of ILDC_expert) containing gender-specific sections and religion-specific statutes from the Indian law (See Table 13) for analysis of religion and gender bias in the judgment prediction task (Section 4). In addition, we consider all 56 annotated cases (ILDC_expert) of (Malik et al., 2021). These 256 (100+100+56) cases form the test dataset for judgment predn.

**judgment predn train dataset**: This is the entire training set of (Malik et al., 2021) (see appendix Section B for details) for prediction. Two randomly chosen cases from this training set are used to train LLMs for judgment predn task. In addition, two annotated cases (out of 56) (ILDC_expert) were used for training of judgment predn + expln task for LLM.

| Prediction only (45 cases) | | | |
|---|---|---|---|
| Method | Macro Precision | Macro Recall | Macro F1 |
| LLM | **0.1629** | **0.7904** | **0.2586** |
| InLegalBERT | 0.1082 | 0.1519 | 0.1220 |
| LEGAL-BERT | 0.1039 | 0.1469 | 0.1185 |
| XLNet | 0.1052 | 0.1543 | 0.1215 |
| Prediction with explanation (45 cases) | | | |
| Method | Macro Precision | Macro Recall | Macro F1 |
| LLM | **0.2794** | **0.4004** | **0.2811** |
| InLegalBERT | 0.1082 | 0.1519 | 0.1220 |
| LEGAL-BERT | 0.1039 | 0.1469 | 0.1185 |
| XLNet | 0.1052 | 0.1543 | 0.1215 |
| Prediction only (245 cases) | | | |
| Method | Macro Precision | Macro Recall | Macro F1 |
| LLM | **0.4243** | **0.5378** | **0.3871** |
| InLegalBERT | 0.3120 | 0.3892 | 0.3131 |
| LEGAL-BERT | 0.3403 | 0.3947 | 0.3243 |
| XLNet | 0.3020 | 0.4369 | 0.3327 |

Table 1: Statute Prediction results. The best values are shown in bold.

## 3 Judgement Prediction

Given a whole case (except the final decision), we consider two types of tasks (i) prediction of judgment, viz. **allowed** (the appellant won the case; label 1) or **dismissed** (the appellant lost the case; label 0) (we call this task judgment predn) and (ii) prediction of judgment along with the gener-

### 3.2 Experimental Setup and Results

As the complete judgments are too long to fit in LLM or non-hierarchical models, in these experiments, we have used the last 512 tokens of the judgments as prescribed by Malik et al. (2021).

**LLM models used:** As mentioned in Section 2.2, we tried many LLM models and finally chose GPT-

3.5 turbo for judgment prediction task as it produced the best results.

**Prompts used:** We used Template 1 (Table 9 in appendix) for `judgment predn + expln` task. This prompt, for in-context training in a *few-shot* setup, shows a case-description, gold standard prediction label (1/0), and gold standard explanation text, and for test, ask the LLM to generate both the prediction and explanations. For only prediction, we have used Template 2 (Table 10 in appendix) which is similar to Template 1 except that it does not have the explanation component in prediction. We also used zero-shot prompting which did not work well.

**Baselines:** We deployed InCaseLaw, InLegalBERT (Paul et al., 2023), XLNet (large), XLNet (base) (Yang et al., 2019) and LEGAL-BERT (Chalkidis et al., 2020). We set the batch size to 16, using Adam (Kingma and Ba, 2014) optimizer, conducted training for 3 epochs on `judgment predn` train dataset, and employed a learning rate of $2e-6$. The remaining hyper-parameters were set to their default values as provided by the HuggingFace library. It is to be noted, that it is a binary classification problem.

**Prediction results:** The results of these experiments are reported in Table 3. We report Macro Precision, Macro Racall and Macro F1 on `judgment predn` test dataset and `judgment predn + expln` test dataset. Note that, **Template 2 prompt** runs on `judgment predn + expln` test dataset without generating explanation (shown as "Prediction only (54 cases)") and **Template 1 prompt** while generating explanations for the prediction (shown as "Prediction with explanation (54 cases)"). We note that when the LLM model is prompted to generate explanations along with the prediction, its performance improves than without explanation. On 54 cases, the best performance is by XLNet while LLM performance is comparable with the other models (in Macro F1). However, LLM lags behind the baselines noticeably in 256 cases. As opposed to statute prediction, no context is provided about the judgment decision. As a result, the LLM inferred the decision from a few in-context training examples. In addition, the fine-tuned baseline models possibly benefited from having a simpler binary classification task than the multi-label statute prediction objective and were, therefore, able to perform better.

**Explanation results:** The results are shown in Table 2 under "Judgment Prediction". ROUGE-

1, ROUGE-2, ROUGE-L, METEOR and BLEU scores of Table 2 are comparable with those reported in Table 5 (Machine explanations v/s Expert explanations) by (Malik et al., 2021), which possibly indicates that the quality of explanations generated by LLMs is not yet of satisfactory level when the verbatim match is considered. However, we see that the BERTScore values are considerably better which possibly indicates that the explanations have good semantic alignment with the gold standard explanations. Table 8 (in appendix) shows some decent explanations generated by LLM in comparison with the expert-labeled annotations and the generated explanations for judgment prediction, in general, are well-aligned with expert-annotated explanations.

**Expert opinion on LLM-generated explanations:** On randomly selected 10 cases from `judgment predn` test dataset (which do not have annotated explanations) the legal expert assigned an average score of 7.2 out of 10. For some of the cases, the scores are as high as 9.5 out of 10. This possibly also accounts for a high BERTScore for `judgment predn + expln` as reported in Table 2.

| Prediction only (54 cases) | | | |
|---|---|---|---|
| Method | Macro Precision | Macro Recall | Macro F1 |
| LLM | 0.5922 | 0.5893 | 0.5850 |
| InCaseLaw | 0.7708 | 0.6071 | 0.5278 |
| XLNet (large) | **0.7286** | **0.6758** | **0.6493** |
| XLNet (base) | 0.6882 | 0.6223 | 0.5786 |
| InLegalBERT | 0.7826 | 0.6429 | 0.5833 |
| LEGAL-BERT | 0.7341 | 0.6414 | 0.5940 |
| Prediction with explanation (54 cases) | | | |
| Method | Macro Precision | Macro Recall | Macro F1 |
| LLM | 0.6534 | 0.6481 | 0.6451 |
| InCaseLaw | 0.7708 | 0.6071 | 0.5278 |
| XLNet (large) | **0.7286** | **0.6758** | **0.6493** |
| XLNet (base) | 0.6882 | 0.6223 | 0.5786 |
| InLegalBERT | 0.7826 | 0.6429 | 0.5833 |
| LEGAL-BERT | 0.7341 | 0.6414 | 0.5940 |
| Prediction only (256 cases) | | | |
| Method | Macro Precision | Macro Recall | Macro F1 |
| LLM | 0.5703 | 0.5688 | 0.5649 |
| InCaseLaw | 0.7131 | 0.6420 | 0.6031 |
| XLNet (large) | **0.7336** | **0.7049** | **0.6913** |
| XLNet (base) | 0.7171 | 0.6786 | 0.6589 |
| InLegalBERT | 0.7436 | 0.6579 | 0.6182 |
| LEGAL-BERT | 0.7221 | 0.6898 | 0.6735 |

Table 3: Judgement Prediction results. The best results are shown in bold.

## 4 Bias, Fairness and Ethics

In this section, we perform tests for the presence of gender and religious bias in the LLM predictions.
**Statute Prediction for religion and gender-specific sections:** On `statute predn + expln` test dataset, the LLM performance *deteriorates* for cases containing religion-specific sections (R) (See Table 14 in appendix) than that in neutral cases. In `statute predn` test dataset, the performance of

LLMs *deteriorates* for cases containing religion-specific (R) or gender-specific (G) cases. Note that we already chose cases containing gender/religion-specific statutes earlier (see Sections 2 and 3).

**Gender frequency disparity:** We have analyzed the number of gender-definition words (Zhao et al., 2018) in the gold-standard and LLM-generated explanations. Figure 3 (in appendix) shows for statute prediction (SP) there is a radical increase in Male:Female ratio in the LLM generated explanations (M/F ratio: 2.62) than that in the expert-annotated explanations (M/F ratio: 1.80). This disparity is less pronounced for judgment prediction (JP) though there is a slight increase. We do not report this for religion due to the lack of a good religion-definition wordlist.

**Group disparity:** We report the **Worst Class Influence** (WCI) score per group (Chalkidis et al., 2022) to note group disparity for the under-represented groups: Female, Muslim in India. Table 15 shows higher values and hence higher disparity for females in statute prediction and for Muslims in judgment prediction.

(please see Appendix Section D for more details)

**Ethical concerns:** We have obtained a note on the ethical concerns of using LLMs in tasks like Statute Prediction and Judgement Prediction from a senior legal expert. For the paucity of space, we have included it in the Appendix Section E.

## 5 Conclusion

In this paper, we applied state-of-the-art LLMs for Statute Prediction and Judgment Prediction. We see that LLMs (i) outperform standard baselines in statute prediction, (ii) underperform similar baselines for judgment prediction, (iii) for a subset of cases, generate explanations moderate for statute prediction, but often quite good for judgment prediction and (iv) exhibits evidence of gender and religion bias in the predictions and generated explanations. The commendable performance in a complex task like multi-label statute prediction (also with explanations) with no explicit training in the legal domain, possibly shows promise in the application of LLMs in future legal tasks. At the same time, judicious monitoring of bias, fairness and ethics in LLMs is a desideratum.

## Limitations

Due to the token limitation and high subscription charges of paid LLMs, we could not do experiments on a larger number of cases. Also, legal expert annotations are expensive and time-consuming to obtain (though, to our knowledge, we have used some of the biggest legal expert annotated datasets available for Indian cases, in this paper). Consequently, the findings reported in this paper regarding LLMs may not hold for the whole Supreme Court collection or for an entire judicial system. We plan to do more rigorous experiments on bigger datasets, as and when available, to ensure the statistical validity of research findings on the application of LLMs in high-stake domains like law. We have not used a baseline model that can jointly predict (statute/judgment) and generate explanations, which could have been an interesting comparison with the LLM that has simultaneously predicted and generated explanations in this paper. We see that statute prediction using LLMs has done better than baselines that do not consider the textual content of the statutes. We have not compared with (Paul et al., 2022) and (Paul et al., 2020) which leverage the text of statutes due to their requirement for additional citation graph information, handcrafted legal hierarchies and their specificity (focusing on criminal law).

## Ethics Statement

The law students who did annotations (upon their consent) were treated fairly for the datasets used in this paper, have been fairly compensated and they are not authors of this paper. Two hundred case documents (100 for statute prediction, 100 for judgment prediction) were chosen such that they contained gender-related or religion-related statutes (see Table 13 for these statutes) to ensure that we have cases to examine existing gender or religion bias and not to incorporate any bias consciously. We have duly paid subscription fees to OpenAI for using their paid GPT models used in this paper.

## Acknowledgement

This work is supported in part by the Science and Engineering Research Board, Department of Science and Technology, Government of India, under Project SRG/2022/001548 and Microsoft Academic Partnership Grant 2023 Agreement No. 7581365. Koustav Rudra is a recipient of the DST-INSPIRE Faculty Fellowship [DST/INSPIRE/04/2021/003055] in the year 2021 under Engineering Sciences.

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

## A Hallucinations

## B Judgment Prediction Dataset

(Malik et al., 2021) presents a meticulously curated collection of case proceedings from the Supreme Court of India (SCI) in the English language. Within the SCI's decision-making process, a judge or bench evaluates claims filed by the appellant/petitioner against the respondent, ultimately determining whether these claims should be "accepted" or "rejected." The ILDC dataset encompasses these case proceedings, providing valuable labeled data with the original decisions rendered by the SCI judges, serving as gold labels for analysis and evaluation.

The ILDC involved publicly available SCI case proceedings spanning from 1947 to April 2020. The ILDC is subdivided into three distinct sets. The ILDC_single subset comprises cases with either a single petition or multiple petitions resulting in the same decision. On the other hand, the ILDC_multi subset encompasses cases involving multiple appeals that lead to differing decisions. Lastly, the ILDC_expert subset consists of annotated case documents where legal experts have provided explanations elucidating the rationale behind the decisions. The experts predict the judgment outcomes and mark specific sentences they consider as explanations, assigning ranks to indicate the relative importance of each sentence in shaping the final judgment. This subset allows for evaluating prediction algorithms for the explainability aspect and highlights the differences between ML-based explainability methods and expert explanations.

## C Judgment Prediction Model Details and Hyper-parameters

We conducted experiments using previously mentioned on (Malik et al., 2021) state-of-the-art models such as XLNet (Yang et al., 2019) and transformers trained on legal corpora, namely Legal BERT (Chalkidis et al., 2020), InLegalBERT (Paul et al., 2023), and InCaseLaw (Paul et al., 2023). To fine-tune these models, we utilized the HuggingFace library (Wolf et al., 2019) and trained them on the ILDC_multi dataset, considering only the last 512 tokens of each document. This choice was based on experimental results, which indicated that using the last 512 tokens achieved the best performance.

The fine-tuning process for all transformer-based models followed a consistent approach. We set the batch size to 1, using Adam (Kingma and Ba, 2014) optimizer, conducted training for 3 epochs, and employed a learning rate of $2e-6$. The remaining hyper-parameters were set to their default values as provided by the HuggingFace library.

By leveraging these transformer architectures and fine-tuning techniques, we aimed to develop accurate case prediction models capable of effectively analyzing the ILDC_multi dataset.

## D Bias and Fairness

**Statute Prediction for religion and gender-specific sections:** We analyzed the performance of LLM on the religion-specific sections (Table 13), where it performed best in 11 cases without explanations. We also observed the performance of LLM in the gender-specific sections in 85 cases without explanation. Overall, LLM performed best for the sections which contained no gender or religion-specific sections in 144 cases without explanation. On `statute predn + expln` test dataset, the LLM performance *deteriorates* for cases containing religion-specific sections (R) (Table 13) than that in neutral cases. In `statute predn` test dataset, the performance of LLMs *deteriorates* for cases containing religion-specific (R) or gender-specific (G) cases (see Table 14).

**Gender frequency disparity:** We have analyzed the number of gender-definition words (Zhao et al., 2018) in the gold-standard and LLM-generated explanations as, we think, the gold-standard explanations annotated by legal experts serve as a focussed summary aligned towards the topic (governing the prediction of statutes or decisions) of the cases.

Figure 3 contains overall word distribution plots for both tasks. 45 fact texts from Indian Supreme Court cases used for the statute prediction task. In the annotated explanations, the ratio of male to female definition words in the annotated explanation was 1.80, and in the generated explanation was 2.62. For 54 annotated cases (ILDC_expert) used for the judgment prediction task, the ratio in male to female definition words in the annotated explanation was 6.21, and in the generated explanation was 6.87. For the statute prediction task, the male to female words ratio in annotated explanation is lower than in generated explanation. Whereas, for the judgment prediction task, the male-to-female ratio in annotated explanation is higher than in generated explanation. Figure 3 shows for statute prediction

| Fact Text |
|---|
| Appellants call in question legality of the judgment rendered by the High Court upholding conviction of the appellants (hereinafter referred to as the accused;) and sentence as imposed by the trial Court which had sentenced each to undergo rigorous imprisonment for three months, two years and seven years respectively with separate fines for each of the alleged offences with default stipulations. Background facts leading to the trial of the accused appellants are as follows: The case was registered on the basis of information lodged by P1 (PW-6), which was recorded on 10.11.1989 at about 2.00 a.m. According to the informant, he and his son P2;s wife P3 (PW-7) were sitting in the courtyard of the house of P2 (hereinafter referred to as the deceased;). It was about 11.00 a.m. on 9.11.1989 when deceased was coming from the village after purchasing vegetables. When he reached near the house of P4, son of P5, P6 (A-1) armed with a Gandasi and P7 (A-2) armed with a lathi were present there. P7 made an obscene gesture. At this P7 and the deceased exchanged hot words and abused each other. P6 gave a Gandasi blow on the right hand of the deceased, which caused a grievous injury. P7 gave a lathi blow on the left foot of the deceased and also gave a thrust blow of lathi on the left side of his head. Deceased fell down on the ground. The occurrence was witnessed by P1 (PW-6) and P3 (PW-7). |
| **Applicable statutes** |
| IPC 34: Acts done by several persons in furtherance of common intention |
| IPC 325: Punishment for voluntarily causing grievous hurt |

Table 4: Example case from statute `statute predn + expln` test dataset. The colors show the explanation annotations and the corresponding statutes are shown in the same colours. We asked law students from a reputed law university to perform these annotations for 45 cases of FIRE AILA 2019 dataset (Task 2: Statute retrieval) (Bhattacharya et al., 2019a). There were two students who independently did the annotations and a senior faculty of law resolved disagreements, if any.

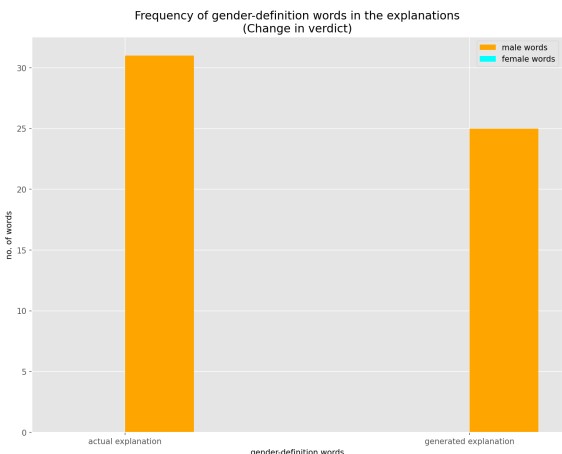

Figure 1: Male:Female ratios in explanations for accurate Judgment Prediction by LLM

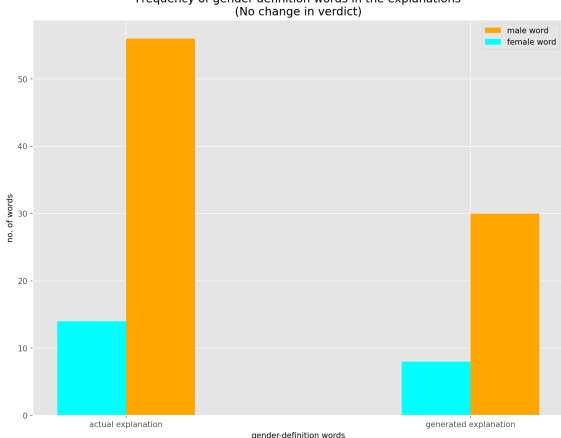

Figure 2: Male:Female ratios in explanations for wrong Judgment Prediction by LLM

(SP) there is a radical increase in male:female ratio in the LLM generated explanations (M/F ratio: 2.62) than that in the expert-annotated explanations (M/F ratio: 1.80). This disparity is less pronounced for judgment prediction (JP) though there is a slight increase. For a particular case, for statute prediction, the expert-annotated explanation read: "*both the deceased persons were killed by the appellant by inflicting dagger blows, P3 and P9 died at the spot.*" while the corresponding LLM-generated explanation read: "*asked the appellant as to why he was stabbing P3. P7 stated that she was the root of all troubles so the appellant started stabbing P9 at her abdomen, neck and other parts of her body.*" In the LLM-generated explanation, apart from factual inaccuracies, we see the injection of one male and three female words into the text. In addition, we see that in cases where there LLM wrongly predicts a verdict, there are no female words (see Figure 2); however there are female words, in lower proportion than male words, for the cases where LLM makes a correct prediction (see Figure 1).

**Group disparity:** The unequal performance across statutes and judgments affects the macro-averaged performance across gender groups (male and female) and religious groups (Hindu and Muslim). We report the **Worst Class Influence** (WCI) score per group (Chalkidis et al., 2022). We observed the under-represented group (Female, Muslim); in the statute prediction task, the Female group is the group with the worst performance (higher values), while in the judgment prediction task, the Muslim group performed worst (See Table 15). The cases that contained male and female definition words (Zhao et al., 2018) in the annotated explanation for the statute prediction or judgment prediction were considered the test cases for those groups. The cases that contained Hindu and Muslim names, as either appellant or respondent of the cases, were considered the test cases for those groups. The Hindu and Muslim names were identified using a surname word list curated by us (which we will release with the paper).

## E Ethical Concerns about LLMs in legal domain: a legal expert perspective

The Large Language Models (LLMs), from what has been seen of them and their work so far, do have tremednous potential and an increasing degree of sophistication when it comes to the manner in which they may be used in the the society in

general and the legal domain in particular. From search engine, voice assistance, preservation of languages and debugging codes to providing research assistance, judgment and statute prediction, document summarisation and trend analysis, the breakthroughs that have been made by LLMs can certainly open up a new vista of possibilities in front of legal professionals as well as the subjects governed and affected by law. At the same time, usage of such technology also carried significant social and ethical challenges and the implications of such challenges may assume particular gravity in the legal context. The emergent capability of LLMs of performing a range of downstream tasks, including tasks requiring an increasing level of intuitive association, processing and decision-making, unless coupled with adequate oversight, guidance, responsible design and operation with humans-in-the-loop, may not only remain relegated to a superficial stage with tremendous potential of spreading misinformation and harmful content at an unprecedented scale and scope, but also have a deep impact on the legal and justice system as a whole. Such impact can be felt via concerns such as distribution of harmful content and solutions generated by LLMs trained on biased and discriminatory data, exacerbating reputational, financial and legal risks of proprietary solutions being used in an unauthorised manner to generate data and strategy options for the users, dislcosure of sensitive information, compromising data privacy and data security, amplification of existing bias or inaccuracy by multiple times due to large scale usage, data hallucination leading to unwanted and harmful outcomes, workforce displacement and other issues. For instance, lawyers having used LLMs to generate supporting case-laws for their arguments have already fallen victim to fake citations and cases being provided. Similarly, if an LLM makes wrong predictions about judicial orders, applies incorrect statutory provisions or fails to apply correct provisions to fact situations while generating an opinion, passes off proprietary or even worse, imaginary solutions as innovative, then the potential harm caused to the legal sector and the justice system can be catastrophic. Having said that, a focus on explainability, sector-specific training, responsible design, and multi-layered audits with adequate human supervision on the design and dissemination stage of LLM technology, on the post-training but pre-release stage of such models and on the usage stage for applications based on

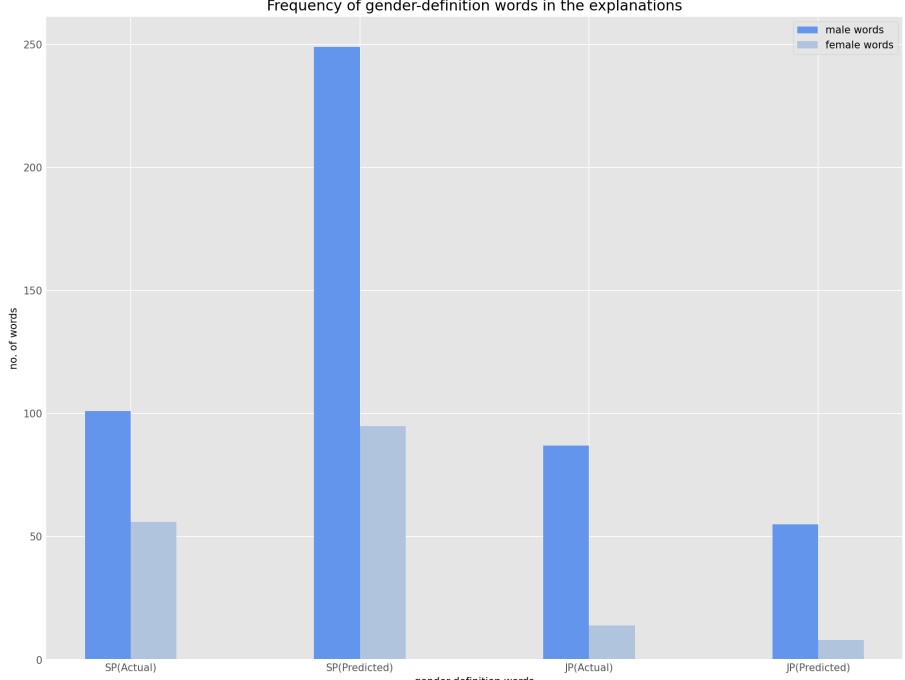

Figure 3: Number of male and female words in the gold-standard explanations and the LLM-generated explanations for Statute Prediction and Judgment Prediction.

LLMs, can actually go a long way to assuage most of these ethical concerns. For instance, if an LLM is used to predict the verdict of the court by going through the rest of the judgment and at the same time, it is also used to generate explanations about which parts of the judgment it has relied upon for such prediction, then the legal expert as a human supervisor will easily be able to identify any potential bias or inaccuracy involved in the prediction. In fact, there is a possibility that such use of LLMs may actually deliver better results than the current practices, because while even human experts will have bias of their own, their conclusions are not always supported with explanations on every level, something which the LLMs can provide owing to their greater processing power and performance speed. In the course of this very research, the legal expert has found the accuracy and explainability of the LLM in judgment prediction to have reached a high level of performance, based on the sampled data review, while the same explainability assisted the expert to identify the drawbacks of the LLM in statute prediction. Filtered data use for training and test purposes coupled with explainability can be used to counter instances of racial, gender, caste or related bias. Similarly, the governance audits referred to above can be perceived or shaped as the training received by even human experts in legal

and professional ethics. The data hallucination aspect can be checked at the model-level audit. The workforce displacement issue can be countered by considering the LLM technology as an augmentative and not a substitutive force for human effort in knowledge retrieval, making decisions and similar cognitive and intuitive processes. Eventually, if such an approach is adopted, the LLMs can evolve into efficient and trustworthy assistive technology for legal information and expertise management, without compromising on the ethical scale.

## F Chain Of Thought prompting

Yu et al., 2022 (Yu et al., 2022) used zero-shot Chain-Of-Thought (CoT) prompting for statute prediction, where they first generated explanation and then used it for prediction. However, this did not produce the best results (than without explanations) and also they did not present any analysis of the quality of explanations. We, however, have integrated prediction and explanation as a part of the prompts (in both templates 1 and 2, for statute prediction, by stating additionally "extract words or lines from the case due to which the predicted statutes are applicable".

For statute prediction, to adhere to the more conventional few-shot CoT prompting (Wei et al., 2022) where the predictions with explanations are

a part of few-shot prompting training, for the rebuttal, we have designed a few-shot prompt for statute prediction as below:

<start of prompt>

prompt = f"""Given a fact statement as a Question delimited by triple backticks ("') and multiple choice options representing the names of statutes, predict the applicable statutes from the Statute Options, and the explanation for each of the applicable statute containing lines from the Question delimited by triple backticks("'). Example is given for your reference. The lecture provides a description of each statute, and your task is to select the statutes from Statute Options that best align with the words from the given Question. Choose the most relevant statutes and lines that match the given context. Statute options (Answer must always be from these) and Lecture (description of statute) are same for each Question. Answer must always be from Statute Options.

Example :

Fact statement: "These appeals are directed against the judgment of a High Court whereby an appeal and a criminal revision were disposed of. The appellants were found guilty and sentenced to undergo various terms of sentences. The Criminal Appeal was filed by three appellants questioning the conviction and sentence as recorded. Complainant filed a revision petition stating that she was entitled to compensation. Background facts giving rise to the trial are essentially as follows: .."

Statute options (Answer must always be from these):

1. "Constitution_226" 2. "Constitution_136" ...

Lecture (description of statute):

1."Constitution_226": Title: Power of High Courts to issue certain writs Description: (1) Notwithstanding anything in Article 32 every High Court shall have powers, throughout the territories in relation to which it exercise jurisdiction, to issue to any person or authority, including in appropriate cases, any Government, within those territories directions, orders or writs, including writs in the nature of habeas corpus, mandamus, prohibitions, quo warranto and certiorari, or any of them, for the enforcement of any of the rights conferred by Part III and for any other purpose

2. "Constitution_136": Title: Special leave to appeal by the Supreme Court Description: (1) Notwithstanding anything in this Chapter, the Supreme Court may, in its discretion, grant spe-

cial leave to appeal from any judgment, decree, determination, sentence or order in any cause or matter passed or made by any court or tribunal in the territory of India. (2) Nothing in clause (1) shall apply to any judgment, determination, sentence or order passed or made by any court or tribunal constituted by or under any law relating to the Armed Forces.

...

Answer : ['Indian Penal Code, 1860_321', 'Indian Penal Code, 1860_324', 'Indian Penal Code, 1860_302', 'Indian Penal Code, 1860_34']

Explanation (words from the fact statement for each of applicable statute):

Statute: "Indian Penal Code, 1860_34" Lines : "the appellants felt offended and when the members of the complainant party came forward and obstructed the appellant from doing the work and restrained them from pulling out the pipe."

Statute: "Indian Penal Code, 1860_321" Lines : " attacked the complainant party. the first injury to be an incised wound. Second and third were abrasions on the left shoulder and neck. The fourth injury was a lacerated wound on the right parietal area of scalp. On the post-mortem conducted on P4, an incised wound was found on the parietal area of the scalp"

Statute: "Indian Penal Code, 1860_324" Lines : " P4 was attacked ...attacked the complainant party. "

#####

Format of Response : Answer containing a list of applicable statutes from the Statute Options above ,and Explanation (containing lines for each of the applicable statutes from the Question) for the Fact Statement delimited by triple backticks ("'). Note: Response must not include Description from Lecture(description of statute) as Explanation. Do not include statute options in your response. Lines must be from the Question delimited by triple backticks ("').

Question: "'{Fact_Statement}"'

""" <end of prompt>

Please note that above prompt provides not only the fact text and statute text, but also the extractive explanations from the fact text to train the LLM. For this experiment, 8 cases were used for training the few-shot prompt, while the remaining 37 were used as test cases. This produced **Macro-Precision: 0.41, Macro-Recall: 0.42 and Macro-F1: 0.36, which outperforms the LLM performance (Tem-**

plate 2) that recorded **Macro-Precision: 0.17, Macro-Recall: 0.68 and Macro-F1: 0.27 and In-LegalBERT (the best-performing baseline) that recorded Macro-Precision: 0.18, Macro-Recall: 0.18 and Macro-F1: 0.17, on the same test set**. However, the quality of explanations generated by this CoT prompt is considerably inferior to those produced by LLM (Template 2). This possibly leads us to the conclusion that few shot CoT-style prompting, although being trained with explanations, does not necessarily produce high-quality explanations for the test set.

Kindly note that for judgement prediction and explanation we have already adhered to the more conventional few-shot CoT prompting (Wei et al., 2022) where the predictions with explanations are a part of few-shot prompting training. Kindly refer to Table 9 in the appendix to note that for a given case, the explanation along with the actual prediction is provided in the few-shot prompt training. To our knowledge, few-shot CoT prompting has not been applied to any legal task yet.

For judgement prediction task, we reversed the order of the prediction label and the explanation in the training prompt (that is first explanation and then prediction label) in the few-shot training and requested the corresponding output from the LLM. We have drawn inspiration from the paper Chain-of-Thought Prompting Elicits Reasoning in Large Language Models. Wei et al. 2022 (Wei et al., 2022), Section 3.3 Ablation Study, Chain of thought after answer, in order to analyse the sequential reasoning embodied in the explanation and its dependence/consequence on the prediction ability of the LLM model beyond just extracting knowledge from the input text.

The output explanations were obtained for 80% and were logically reasoned. But, still, only 20% was predicted correctly by the model. This therefore led to a deterioration in the performance of our previous prompt style where prediction **preceded** explanation.

Therefore, from these sets of experimentations, we can safely say that the prompting style used by us previously in the paper was already in alignment with the state-of-the-art style (like CoT).

## G  More baselines

A possible reason for the apparent underperformance of the non-LLM models in statute prediction can be ascribed to the fact that the LLMs use con-

text of the statutes (in Template 2). To ensure this for the baselines, we have designed a transformer, viz. BERT base model (uncased) model[3] that takes into account both the fact text and statute (label) text. In the context of the paired sentence task, the 512 token-limit was shared equally between the facts and statutes, with each segment being allowed a maximum of 256 tokens. The two texts were separated by the [SEP] token as is recognized by BERT for such tasks. Taking up to 150 positive pairs and an equal number of negative pairs for each statute yielded around 22k training samples. For the test set, the pairs were created between a fact text and each statute text. The framework leveraged the BERT base model (uncased) for effectuating binary classification, discerning between the fact text and individual statute texts. The model underwent training over 100 epochs, with the most optimal outcomes observed at the 29th epoch. Note that the binary results for each test fact, statute pair were converted to a multi-label setup (used in our experiments as reported in Table 1), where for each test fact we have the prediction for each of the candidate statutes.

The model produced a **Macro-Precision: 0.07, Macro-Recall: 0.28, Macro-f1: 0.10** on 245 test cases. On 45 test cases (used for prediction and explanation), the results are **Macro-Precision: 0.02, Macro-Recall: 0.04, Macro-F1: 0.03**. Clearly, these results are inferior to the LLM results in Table 1. This can possibly be attributed to the partial context captured by a BERT model due to the token limitation. To alleviate this issue further, we replaced BERT (512 token limit) with Longformer[4] which has a token limit of 4096. After running for 26 epochs (72+ hours), the model produced a **Macro-Precision: 0.05, Macro-Recall: 0.20, Macro-F1: 0.06** on 245 test cases (that is worse than the reported LLM results and also worse than BERT). It is to be noted that Longformer is extremely memory intensive and takes a long time to train.

We also used a Hierarchical Attention Network (HAN) (Yang et al., 2016) which is supposed to consider the complete context of the text and which produced **Macro-precision = 0.05 Macro Recall = 0.31 and Macro-F1 = 0.06**, on 245 cases. For both models, we observed steady training and no signs of overfitting.

---

[3]https://huggingface.co/bert-base-uncased
[4]https://huggingface.co/docs/transformers/model_doc/longformer

This possibly shows that the paired-text task is challenging for a multi label-setup with 80 classes and also upholds the efficacy of the LLM prompts used in the paper.

## H    Additional experiments

For template 2, we have also used statute numbers like "Indian Penal Code Section 1860_5", "Special Courts Act, 1979_5", "Constitution Article 136" etc. instead of statute ids like S1, S2 etc. However, the prediction results for 45 cases (Macro precision: 0.11, Macro recall: 0.56, Macro F1: 0.18), were worse than the original Template 2 results and so was the explanation quality. So, we can possibly infer that providing the actual statute numbers does not necessarily help the task.

## I    Data Contamination

We checked the paper Language Models are Few-Shot Learners, Brown et al. 2020 (Brown et al., 2020)), and found on pages 8-9 (Table 2.2) that the datasets on which GPT-3 is trained are Common Crawl (filtered), WebText2, Wikipedia etc. and we found no documentation (here or in other sources like https://help.openai.com/en/articles/7842364-how-chatgpt-and-our-language-models-are-developed) stating whether Indian Supreme Court Judgements (on which we have reported our results) belong to this training set. So, we have not identified any evidence of data contamination, i.e., whether GPT has already seen the test cases on which the tasks are reported in this paper.

| Template 1 (only prediction) |
|---|
| **Task:** Given examples of a Supreme Court case and the statutes applied in that case, your objective is to make accurate predictions of the specific charge or statute that is most likely to be applied within the context of the case delimited by triple backticks (""'), ensuring exact predictions and learning from the provided examples. You should only include the statutes it is most confident about. The response format should include the statutes applied as in the context. You should showcase creativity and knowledge to enhance the accuracy of statute predictions based on the given fact statement. 
 **Training:** 
 Fact Statement: Statutes: 
 ### 
 Fact Statement: Statutes: 
 **Response and instructions:** 
 Format your response as follows: Statutes applied: [List of applicable statutes] 
 Learn from the examples provided in the context to understand the task of charge or statute prediction. 
 Your response should be focused on providing the exact statute or charge that aligns with the legal principles and precedents applicable to the given facts. 
 In your response, include only the statutes you are most confident about. 
 Ensure that the statutes generated as responses are valid and recognized legal statutes. Avoid generating fabricated or invalid statutes. The model's performance will be evaluated based on its ability to predict the correct statute, include only confident statutes, and showcase creativity in its predictions. |
| **Template 1 (prediction + explanation)** |
| (For Explanations these lines were included in the prompt:) |
| . . . and extract words or lines from the case due to which the predicted statutes are applicable, 
 . . . You should only include the statutes and words or lines from the fact statement you are most confident about. 
 . . . Format your response as follows: 
 Statutes applied: [List of applicable statutes] 
 Explanation:[[Statute:words or lines from the fact statement due to which the statute is applicable]] 
 . . . Note: Words or lines must be from Fact Statement delimited by triple backticks("""). |

Table 5: Prompts in Statute Prediction (Template 1)

| Template 2 (only prediction) |
|---|
| **Task:** |
| You are given a fact statement delimited by triple backticks ("") and statutes with their title and description. Your task is to identify the statutes applicable to the fact statement from the given statutes that you are most confident apply to the fact statement. Each statute consists of a title and a description of its scope and provisions. Include only those statute in your response which description logically matches with some parts of the fact statement. |
| **Training:** |
| Statutes: |
| Statute Id: |
| Title: |
| Description: |
| ### |
| Statute Id: |
| Title: |
| Description: |
| **Response and Instructions:** |
| Format of response:[Python list of applicable statutes from Statutes] |
| Your response should include the statutes applicable to the fact statement. The applicable statute must be mentioned exactly as it appears in Statutes provided. Include only those statutes which you are very sure about. |
| **Template 2 (prediction + explanation)** |
| For Explanations these lines were added: |
| . . . along with the words or lines from the fact statement due to which the statute is applicable for each of the statute applicable. |
| . . . The applicable statute should be mentioned exactly as it appears in the list, and the words should be from the fact statement. |
| . . . Explanation : [words/lines from the fact statement due to statute is applicable]. |

Table 6: Prompts in Statute Prediction (Template 2). Temperature = 0, max_tokens between 40 and 500 was used.

| Annotated explanation | Generated explanation |
|---|---|
| In the fight, the accused persons started inflicting injuries to the complainant, and his sons P5 (A-2) gave a pharsi blow on the head of P2, P4 (A-1) caused injury to P1 (PW-5) with two jaily blows. Additionally, P7 and P6 attacked the complainant with lathi blows on shoulder and left elbow respectively and caused several other injuries to the complainant party. | In the fight, the accused persons started inflicting injuries to the complainant, and his sons P5 (A-2) gave a pharsi blow on the head of P2, P4 (A-1) caused injury to P1 (PW-5) with two jaily blows. Additionally, P7 and P6 attacked the complainant with lathi blows on shoulder |
| ..the first injury to be an incised wound. Second and third were abrasions on the left shoulder and neck. The fourth injury was a lacerated wound on the right parietal area of scalp. On the post-mortem conducted on P4, an incised wound was found on the parietal area of the scalp. | Punishment for voluntarily causing hurt "On examining P6, he found the first injury to be an incised wound. Second and third were abrasions on the left shoulder and neck. The fourth injury was a lacerated wound on the right parietal area of scalp." |
| …they conspired with stamp vendors and document writers and other staff to gain monetary benefit and resorted to manipulation of registers and got the registration of the documents with old value of the properties, resulting in wrongful gain to themselves and loss to the Government, and thereby cheated the public and the Government. | The main allegation against the respondents was that while they were working as Sub-Registrars in various offices in the State, they conspired with stamp vendors and document writers and other staff to gain monetary benefit and resorted to manipulation of registers and got the registration of the documents with old value of the properties, resulting in wrongful gain to themselves and loss to the Government, and thereby cheated the public and the Government. |

Table 7: Competent explanations (with statute prediction) generated by LLM. We see that the LLM-generated explanations are very close to legal-expert annotated explanations.

| Annotated explanation | Generated explanation |
|---|---|
| On a full review of the facts it is clear that this transaction was voluntarily entered into in order indirectly to facilitate the carrying on of the business of the company and was made on the ground of commercial expediency. It therefore falls within the purview of section 12(2) of the Income-tax Act, 1922 before its amendment in 1939. This being an investment company, if it borrowed money and utilised the same for its investments on which it earned income, the interest paid by it on the loans will clearly be a permissible deduction under section 12(2) of the Income- tax Act. Whether the loan is taken on an overdraft, or is a fixed deposit or on a debenture makes no difference in law. | On a full review of the facts it is clear that this transaction was voluntarily entered into in order indirectly to facilitate the carrying on of the business of the company and was made on the ground of commercial expediency. It therefore falls within the purview of section 12(2) of the Income-tax Act, 1922 before its amendment in 1939. This being an investment company, if it borrowed money and utilised the same for its investments on which it earned income, the interest paid by it on the loans will clearly be a permissible deduction under section 12(2) of the Income-tax Act. Whether the loan is taken on an overdraft, or is a fixed deposit or on a debenture makes no difference in law. The only argument urged against allowing this deduction to be made is that the person who took the debentures was the party who sold the ordinary shares. It cannot be disputed that if the debentures were held by a third party, the interest payable on the same would be an allowable deduction in calculating the total income of the assessee company. What difference does it make if the holder of the debentures is a shareholder. |
| we feel no doubt or difficulty in reaching the conclusion that the intervention of the Industrial Tribunals was intended to be excluded and the matter was intended to be kept within the discretion of the Central Government so far as the payment of bonus by the insurers is concerned. Having regard to the unqualified and absolute prohibition contained in s. 31A(1)(c) it seems to us difficult to hold that the payment of bonus to the employees of Insurance Companies is not absolutely conditioned by proviso (vii). In the absence of the said provision no bonus could have been claimed by Insurance employees, and so the effect of the said provision must be to limit the said right to the conditions prescribed by it. | The intervention of the Industrial Tribunals was intended to be excluded and the matter was intended to be kept within the discretion of the Central Government so far as the payment of bonus by the insurers is concerned. Having regard to the unqualified and absolute prohibition contained in s. 31A(1)(c) it seems to us difficult to hold that the payment of bonus to the employees of Insurance Companies is not absolutely conditioned by proviso (vii). In the absence of the said provision no bonus could have been claimed by Insurance employees, and so the effect of the said provision must be to limit the said right to the conditions prescribed by it. That is why we think that the Tribunal was right in coming to the conclusion that the reference made by the Central Government is invalid. |

Table 8: Competent explanations (with judgment prediction) generated by LLM. We see that the LLM-generated explanations are very close to legal-expert annotated explanations.

| Template 1 (prediction + explanation) |
|---|
| **prompt** = f“““Task: Given a Supreme Court of India case proceeding enclosed in angle brackets $<>$, your task is to predict the decision of the case (with respect to the appelant) and provide an explaination for the decision.
**Prediction**: Given a case proceeding, the task is to predict the decision 0 or 1, where the label 1 corresponds to the acceptance of the appeal/petition of the appellant/petitioner and the label 0 corresponds to the rejection of the appeal/petition of the appellant/petitioner, Explanation: The task is to explain how you arrived at the decision by predicting important sentences that lead to the decision.
**Context**: Answer in a consistent style as shown in the following two examples:
**case_proceeding**: # case_proceeding example 1
**Prediction**: # example 1 prediction
**Explanation**: # example 1 explanation
**case_proceeding**: # case_proceeding example 2
**Prediction**: # example 2 prediction
**Explanation**: # example 2 explanation
**Instructions**: Learn from the above given two examples and perform the task for the following case proceeding.
case_proceeding: $<${case_proceeding}$>$
Format your output in list format: [prediction, explanation]”””” |

Table 9: Prompts in Judgment Prediction (Template 1: prediction + explanation). Temperature = 0 was used.

| Template 2 (prediction only) |
|---|
| **prompt** = f“““Task: Given a Supreme Court of India case proceeding enclosed in angle brackets $<>$, your task is to predict the decision of the case (with respect to the appellant).
**Prediction**: Given a case proceeding, the task is to predict the decision 0 or 1, where the label 1 corresponds to the acceptance of the appeal/petition of the appellant/petitioner and the label 0 corresponds to the rejection of the appeal/petition of the appellant/petitioner
**Context**: Answer in a consistent style as shown in the following two examples:
**case_proceeding**: # case_proceeding example 1
**Prediction**: # example 1 prediction
**case_proceeding**: # case_proceeding example 2
**Prediction**: # example 2 prediction
**Instructions**: Learn from the above given two examples and perform the task for the following case proceeding.
**case_proceeding**: $<${case_proceeding}$>$
Give the output predicted case decision as either 0 or 1.”””” |

Table 10: Prompts in Judgment Prediction (Template 2: prediction only). Temperature = 0 was used.

| Case description | Actual statutes/explanation | Predictions with Hallucinations | Observation |
|---|---|---|---|
| Sexual assault case | 'Constitution_226', 'Constitution_14', 'Constitution_21', 'Constitution_32', 'Indian Penal Code, 1860_376' | 'Constitution_226', 'Constitution_136', 'Indian Penal Code 1860_376', 'Indian Penal Code 1860_376A', 'Indian Penal Code 1860_498A', 'Indian Penal Code, 1860_302', 'Indian Penal Code 1860_323', 'Indian Penal Code 1860_504', 'Indian Penal Code 1860_506', 'Indian Penal Code 1860_34', 'Indian Penal Code 1860_307', 'Indian Penal Code 1860_467', 'Indian Penal Code 1860_471', 'Indian Penal Code 1860_406', 'Indian Penal Code 1860_468', 'Indian Penal Code 1860_2', 'Indian Penal Code 1860_409', 'Code of Criminal Procedure 1973_482' | The LLM model tends to incorporate statutes that are not relevant to the context; e.g. the case was not on murder and yet the section for murder like 'Indian Penal Code, 1860_302' has been predicted (the correct predictions are in blue) |
| Case on appeal for special leave, an excerpt is given as follows: "the respondent that the imported goods fell within this item and were liable to be charged with duty at that rate.**The Custom authorities**, however, considered that the consignment fell within the description articles plated with gold or silver being item on which duty was payable at per cent.. adjudicated the duty on this latter basis and thereafter the respondent having filed an appeal to [ORG], the levy was upheld by order dated .....The learned Judge further held that the interpretation that he placed upon item in the context of the other entries in the **Tariff** can only be one and it is not reasonably possible for any person to take a contrary view..." | 'Constitution_226', 'Constitution_136' | 'Customs Tariff Act, 1975_25' | The model tends to consider any statute mentioned in the fact statement as the applicable statute in its response, regardless of its actual relevance or applicability. E.g. here the presence of the terms **custom, tariff** led it to wrongly suggest 'Customs Tariff Act, 1975_25' as the applicable statute |

Table 11: Examples of hallucination in Statute Prediction

| Case description | Actual statutes/explanation | Predictions with Hallucinations | Observation |
|---|---|---|---|
| **The following excerpt from a case contains expert annotations marked:** ... The deceased P1 was daughter of PW1. P2 resident of village Sanjkheda and she was married to appellant no. 1 P3 son of P4 about two and half years prior to the date of incident which took place on 15 September 1991. ...For about six months P1 was treated well but thereafter the accused started asking her to bring Rs. 1,000-1,200 from her parents to meet the household expenses and also for purchasing manure. Whenever P1 went to her parental home, she used to tell her parents that her husband and mother-in-law (accused appellants) were harassing her and used to occasionally beat her. Her father PW.1 P2 along with some of his relatives went to the house of the accused and tried to persuade them not to ill-treat P1. Thereafter, the accused treated P1 properly but after about four months they again started harassing her. A few days before Nag Panchami festival P1 came to her parental home and complained that the accused were not giving her proper food, clothing and even footwear. ... PW.1 then immediately went to the house of the accused along with some of his relatives. There he saw that P1 was lying dead and froth was coming out of her mouth which indicated that she had consumed some poisonous substance. The Police Patil of the village PW.3 P6 lodged an accidental death report at 9.00 p.m. on 15 September 1991 at the police station. ... | Section 2 in the Dowry Prohibition Act, 1961[1], Section 304B in The Indian Penal Code (Dowry death[2] | **LLM statute predictions:** 'Constitution_226', 'Constitution_32', 'Indian Penal Code, 1860_149', 'Indian Penal Code, 1860_307', 'Indian Penal Code, 1860_321', Constitution_32: "The right to move the Supreme Court by appropriate proceedings for the enforcement of the rights conferred by this Part is guaranteed." **LLM explanations for the predictions:** Constitution_226: "This appeal, by special leave, has been preferred against the judgment and order dated 23 February 2005 of the High Court (Aurangabad Bench)" Indian Penal Code, 1860_149: "If an offence is committed by any member of an unlawful assembly in prosecution of the common object of that assembly". Indian Penal Code, 1860_307: Attempt to murder - "Whoever does any act with such intention or knowledge, and under such circumstances that, if he by that act caused death, he would be guilty or murder" Indian Penal Code, 1860_321 "Whoever, except in the case provided for by section 334, voluntarily causes hurt, shall be punished with imprisonment of either description for a term which may extend to one year" | The LLM predictions and the explanations are based on superficial features rather than in-depth issues of the case. E.g. it fails to identify that it is a dowry death case. |

Table 12: Examples of hallucination in Statute Prediction (contd.)

| Religion specific sections | |
|---|---|
| **Statutes** | **Description** |
| IPC Sections 295 –298 | Of Offences relating to Religion |
| IPC Section 153A | Promoting enmity between different groups on grounds of religion, race, place of birth, residence, language, etc., and doing acts prejudicial to maintenance of harmony. |
| **Tentatively religion-specific sections** | |
| **Statutes** | **Description** |
| Act 15 of 2000 | Prevention of Terrorism Act, 2002 |
| Arms act, 1959 | Act of the Parliament of India to consolidate and amend the law relating to arms and ammunition in order to curb illegal weapons and violence stemming from them |
| IPC Sections 120A and 120B | Criminal conspiracy |
| IPC Sections 121–130 | Of Offences against the State |
| IPC Sections 141 – 153 | Of Offences against the Public Tranquillity |
| **Gender-specific sections** | |
| **Statutes** | **Description** |
| IPC Sections 375–377 | Sexual Offences including Rape and Sodomy |
| IPC Sections 493–498 | Of Offences related to marriage |
| IPC Sections 498A | Of Cruelty by Husband or Relatives of Husband |
| IPC Section 304B | Dowry death |

Table 13: Statutes related to religion and gender issues. Note that the "Tentatively religion-specific sections" are not directly related to religion, but are often associated with crimes committed by members of a certain religion.

| Data | Precision | Recall | F1 | No of cases |
|---|---|---|---|---|
| **prediction+explanation on statute predn + expln test dataset (45 cases) (R)** | 0.0722 | 0.0629 | 0.0570 | 11 |
| **prediction only statute predn + expln test dataset (45 cases) (R)** | 0.1055 | 0.2222 | 0.1353 | 11 |
| **prediction+explanation on statute predn + expln test dataset (45 cases) (N)** | **0.2804** | **0.3922** | **0.2913** | 34 |
| **prediction only statute predn + expln test dataset (45 cases) (N)** | 0.1874 | 0.7622 | 0.2858 | 34 |
| **prediction only statute predn test dataset (245 cases) (R)** | 0.0439 | 0.0612 | 0.0465 | 50 |
| **prediction only statute predn test dataset (245 cases) (G)** | 0.1116 | 0.1327 | 0.1182 | 85 |
| **prediction only statute predn test dataset (245 cases) (N)** | **0.4185** | **0.5598** | **0.3860** | 144 |

Table 14: Statute Prediction results for LLM for cases containing religion and gender-specific sections. R: Cases with Religion-specific sections, G: Cases with Gender-specific sections, N: Cases with Neutral sections. The LLMs on N cases outperform those on R and G cases.

| Group | Statute Prediction | Judgment Prediction |
|---|---|---|
| **Male** | 0.06 | 0.60 |
| **Female** | 0.11 | 0.33 |
| **Hindu** | - | 0.44 |
| **Muslim** | - | 0.75 |

Table 15: Worst Class Influence (WCI) score per group. The higher the value, the more the disparity. There are no values for Hindu/Muslim in Statute Prediction because there were no religion-specific statutes in the statute predn + expln test dataset.

Table 16: Top 5 cases from Explanation Generation Analysis (on 45 cases)

| Case Number | Actual Statute Involved | Type of actual statute | Avg of Eval Metric Scores |
|---|---|---|---|
| 34 | ['Constitution_226', 'Constitution_21'] | constitutional right of HC to issue writs, protection of life and personal liberty | 0.7982645277 |
| 43 | ['Indian Penal Code, 1860_120'] | criminal conspiracy | 0.756938956 |
| 3 | ['Indian Penal Code, 1860_321', 'Indian Penal Code, 1860_34', 'Indian Penal Code, 1860_307', 'Indian Penal Code, 1860_324', 'Indian Penal Code, 1860_325'] | voluntarily causing hurt with weapons, actions done with intention, attempt to murder | 0.7340507586 |
| 18 | ['Constitution_226', 'Constitution_14'] | constitutional right of HC to issue writs, equal protection of laws | 0.6652348341 |
| 16 | ['Indian Penal Code, 1860_148', 'Indian Penal Code, 1860_147', 'Indian Penal Code, 1860_149', 'Indian Penal Code, 1860_302'] | rioting, armed with deadly weapons, murder | 0.6506418727 |

Table 17: Worst 5 cases from Explanation Generation Analysis (on 45 cases)

| Case Number | Actual Statute Involved | Type of actual statute | Avg of Eval Metric Scores |
|---|---|---|---|
| 41 | ['Indian Penal Code, 1860_302'] | punishment for committing murder | 0.08537873928 |
| 12 | ['Indian Penal Code, 1860_120', 'Indian Penal Code, 1860_34', 'Indian Penal Code, 1860_302'] | punishment of criminal conspiracy, causing hurt intentionally | 0.08882890817 |
| 26 | ['Constitution_21', 'Constitution_14', 'Indian Penal Code, 1860_302'] | protection of life and personal liberty, equality before law, punishment for murder | 0.1075970114 |
| 17 | ['Indian Penal Code, 1860_307', 'Constitution_136', 'Indian Penal Code, 1860_302', 'Indian Penal Code, 1860_34'] | attempt to murder, criminal act done by several person in furtherance of the common intention | 0.1193747665 |
| 32 | ['Constitution_14', 'Constitution_21'] | protection of life and personal liberty, equality before law | 0.1297315706 |

Table 18: Top 5 Performing Statutes from Prediction Analysis (on 245 cases)

| Actual Statute | Type of actual statute | Statute wise F1 Score |
|---|---|---|
| Indian Penal Code, 1860_302 | offences affecting life including murder, culpable homicide | 0.8253741981 |
| Indian Penal Code, 1860_364 | kidnapping or abducting in order to murder | 0.7958333333 |
| Indian Penal Code, 1860_366 | kidnapping, abducting or inducing woman to compel her for marriage, etc. | 0.7914893617 |
| Indian Penal Code, 1860_306 | abetment of suicide | 0.7826086957 |
| Indian Penal Code, 1860_376 | sexual offences including rape, etc. | 0.7402409639 |

Table 19: Worst 5 Performing Statutes from Prediction
Analysis (on 245 cases)

| Actual Statute | Type of actual statute | Statute wise F1 Score |
|---|---|---|
| Indian Penal Code, 1860_467 | offences relating to documents- forgery of a valuable security, will, etc. | 0.4524265645 |
| Indian Penal Code, 1860_409 | offences against property-criminal Breach of Trust | 0.4556451613 |
| Indian Penal Code, 1860_2 | introduction-punishment of offences committed within India | 0.4738867955 |
| Constitution_142 | article providing unique power to the Supreme Court, to do complete justice between the parties, where, at times, the law or statute may not provide a remedy. | 0.4739076155 |
| Arms Act, 1959_25 | punishment for possession or carrying of prohibited arms or prohibited ammunition etc. | 0.4776119403 |