# OpenReview forum: "LLMs -- the Good, the Bad or the Indispensable?: A Use Case on Legal Statute Prediction and Legal Judgment Prediction on Indian Court Cases"
_EMNLP/2023/Conference — EMNLP 2023 Findings_

### Official Review · Reviewer_8zfw · 2023-07-24

**Soundness:** 3

**Excitement:**

4: Strong: This paper deepens the understanding of some phenomenon or lowers the barriers to an existing research direction.

**Paper Topic And Main Contributions:**

The paper explores LLM's ability on two tasks namely statute prediction and judgement prediction of Indian Supreme Court cases. They also evaluate LLM's from explainability perspective on those tasks and compare them with expert generated rationales for both the tasks. While judgement prediction + explanation dataset is derived prior work of Malik et al. 2021, they enhanced statute dataset of Paheli et al. 2019 for statute prediction with a legal-expert-explanations. They also obtain separate set of cases dealing with religion and gender specific cases and observed disparities in performance and bias exhibited by LLM's. Their findings reveal that LLMs  outperform in statute prediction and underperform in judgment prediction. LLM's explanation are moderate for statute prediction and often good for judgment prediction. Finally, LLM display gender and religion bias explanations and prediction performance disparities.

**Questions For The Authors:**

A.  I assume in all places training LLM refers to few-shot examples provided to LLM when prompting. Please clarify this terminology and refine the terminology. (Lien 104, 191, 209)

B. Statute prediction : Line 136-141: In template 2, do you provide any few-shot examples  (Examples of fact statement and statutes?) It only provides statutes and their description ? In that case, why are they called as training in the prompt ?

C. Statute prediction : If LLM for statute prediction are provided with the information about text of statutes, why not employ statute based baselines to compare, rather than fact text only bases? Better baselines would be to consider label text during training to obtain label-aware attention sort of, for fair comparison with LLM's. It would be also interesting to observe what LLM produces if just given statute id without description. It can test whether LLM's already have knowledge of Indian legal statutes.

D. Statute prediction Explanation results - Why not prompt LLM to generate explanation which are extractive in nature rather than abstractive? This might have made evaluation more easier and to draw better conclusions. Though BERT Scores have been shown in Legal summarization works (Shukla et al. 2022) not to reflect true score given they are not trained on legal text specifically.

E. Judgement Prediction: Why are few-shot examples provided in this setting ? (which is mentioned as training and I assume them to be few-shot examples) Did you try with zero-shot way and they did not perform well ? Any specific reason to consider few-shot in this case and not in statute prediction case?

F. Line 248-250: Do you mean judgement prediction instead of statute prediction ?

G. I assume Malik et al. 2021 Dataset of judgement prediction has extractive explanations. Why not prompt LLM for extractive explanations ?

**Reasons To Accept:**

The paper is a interesting direction to explore capabilities of LLM in two specific legal tasks related to Indian Supreme Court cases and establish certain evidences on their performance.

**Reasons To Reject:**

On the whole, some more analysis on generated explanations or different types of hallucinations etc can make it more of an interesting read. I am skeptical if the findings hold true given data contamination issue if the LLM's have already seen the cases etc (unless some probing is carried out to back these claims).

**Reproducibility:**

5: Could easily reproduce the results.

**Reviewer Confidence:**

5: Positive that my evaluation is correct. I read the paper very carefully and I am very familiar with related work.

**Typos Grammar Style And Presentation Improvements:**

Line 28-29: Please use \citealt for better formatting (Yu et al., 2022)(Nguyen et al.,2023)

Line 70-77 : Please specifically mention how annotations are obtained? phrase-level or sentence level ? in-text annotations or free text explanations ? (Table 4 in appendix shows them, please qualify them in writing)

---

> ### Author Rebuttal · Authors · 2023-08-29
>
> We thank the reviewer for the valuable comments. We have tried our best to address them and hope that now the reviewer will have a better understanding of the paper.
>
> A. I assume in all places training LLM refers to few-shot examples provided to LLM when prompting. Please clarify this terminology and refine the terminology. (Lien 104, 191, 209)
>
>
> Response: We have indeed used few-shot where some examples were shown to the LLM, and based on which a prediction was sought for a test case. We will clarify this in the camera-ready version, if accepted.
>
> B. Statute prediction : Line 136-141: In template 2, do you provide any few-shot examples (Examples of fact statement and statutes?) It only provides statutes and their description ? In that case, why are they called as training in the prompt ?
>
> Response: We apologize for the confusion. In fact, for template 2 there was no training (training was for Template 1). We will correct this in the camera ready, if accepted.
>
>
> C. Statute prediction : If LLM for statute prediction are provided with the information about text of statutes, why not employ statute based baselines to compare, rather than fact text only bases? Better baselines would be to consider label text during training to obtain label-aware attention sort of, for fair comparison with LLM's.
>
> Response:
>
> We thank the reviewer for the comment. For the rebuttal, we have now covered this by designing a transformer, viz. BERT base model (uncased) model that takes into account both the fact text and statute (label) text. In the context of the paired sentence task, the 512 token limit was shared equally between the facts and statutes, with each segment being allowed a maximum of 256 tokens. The two texts were separated by the [SEP] token as is recognized by BERT for such tasks. Taking upto 150 positive pairs and an equal number of negative pairs for each statute yielded around 22k training samples. For the test set, the pairs were created between a fact text and each statute text. The framework leveraged the BERT base model (uncased) for effectuating binary classification, discerning between the fact text and individual statute texts. The model underwent training over 100 epochs, with the most optimal outcomes observed at the 29th epoch. Note that the binary results for each test fact, statute pair was converted to a multi-label setup (used in our experiments as reported in Table 1), where for each test fact we have the prediction for each of the candidate statutes.
>
> The model produced a *macro-precision: 0.07,  macro-recall:  0.28, macro-f1: 0.10 on 245 test cases. On 45 test cases (used for prediction and explanation), the results are macro-precision: 0.02, macro-recall: 0.04,  macro-f1: 0.03*. Clearly, these results are inferior to the LLM results in Table 1. This can possibly ascribed to the partial context captured by a BERT model due to the token limitation. In the future, we will use more advanced models for the fact-statute paired text prediction task.
>
> C: It would be also interesting to observe what LLM produces if just given statute id without description. It can test whether LLM's already have knowledge of Indian legal statutes.
>
>
> Response: The statute ids are usually like S1, S2 etc. which do not contain any context of the actual statute used (e.g. Indian Penal Code 302). That is why using the textual description of the statute (as used in Template 2) proved to be an effective strategy.
>
>
> D. Statute prediction Explanation results - Why not prompt LLM to generate explanation which are extractive in nature rather than abstractive? This might have made evaluation more easier and to draw better conclusions. Though BERT Scores have been shown in Legal summarization works (Shukla et al. 2022) not to reflect true score given they are not trained on legal text specifically.
>
>
> Response:
>
> Though we did not explicitly state the LLM to produce extractive summary (we said in the prompt “Your response should include…….along with the words or lines from the fact statement due to which the statute is applicable for each of the statute applicable”), the LLM did produce extractive explanations from within the fact text on most of the occasions. Kindly refer to Table 7 in the appendix where some annotated and LLM-generated explanations are shown. E.g.
>
> while the following was the annotated explanation:
>
> “In the fight, the accused persons started inflicting injuries to the complainant, and his sons P5 (A-2) gave a pharsi blow on the head of P2, P4 (A-1) caused injury to P1 (PW-5) with two jaily blows. Additionally, P7 and P6 attacked the complainant with lathi blows on shoulder and left elbow respectively and caused several other injuries to the complainant party.”
>
> the generated explanation was:
>
> “In the fight, the accused persons started inflicting injuries to the complainant, and his sons P5 (A-2) gave a pharsi blow on the head of P2, P4 (A-1) caused injury to P1 (PW-5) with two jaily blows. Additionally, P7 and P6 attacked the complainant with lathi blows on shoulder”
>
> For statute prediction, to adhere to the more conventional few-shot CoT prompting (Chain-of-Thought Prompting Elicits Reasoning in Large Language Models. Wei et al. 2022) where the predictions with explanations are a part of few-shot prompting training, for the rebuttal, we have designed a few-shot prompt for statute prediction as below. Here we have directed the LLM in the prompt to specifically extract explanations from the fact text. This did not necessarily produce better results as although the explanations were always from the fact text, they did not match well with the gold standard annotations.
>
> <start of prompt>
>
> prompt = f"""Given a fact statement as a Question delimited by triple backticks (```) and multiple choice options representing the names of statutes, predict the applicable statutes from the Statute Options, and the explanation for each of the applicable statute containing lines from the Question delimited by triple backticks(```). Example is given for your reference. The lecture provides a description of each statute, and your task is to select the statutes from Statute Options that best align with the words from the given Question. Choose the most relevant statutes and lines that match the given context. Statute options (Answer must always be from these) and Lecture (description of statute) are same for each Question. Answer must always be from Statute Options.
>
> Example :
>
> Fact statement: "These appeals are directed against the judgment of a High Court whereby an appeal and a criminal revision were disposed of. The appellants were found guilty and sentenced to undergo various terms of sentences. The Criminal Appeal was filed by three appellants questioning the conviction and sentence as recorded. Complainant filed a revision petition stating that she was entitled to compensation. Background facts giving rise to the trial are essentially as follows: .."
>
> Statute options (Answer must always be from these):
>
> 1. "Constitution_226"
> 2. "Constitution_136"
> ...
>
>
> Lecture (description of statute):
>
> 1."Constitution_226":
> Title: Power of High Courts to issue certain writs
> Description: (1) Notwithstanding anything in Article 32 every High Court shall have powers, throughout the territories in relation to which it exercise jurisdiction, to issue to any person or authority, including in appropriate cases, any Government, within those territories directions, orders or writs, including writs in the nature of habeas corpus, mandamus, prohibitions, quo warranto and certiorari, or any of them, for the enforcement of any of the rights conferred by Part III and for any other purpose
>
> 2. "Constitution_136":
> Title: Special leave to appeal by the Supreme Court
> Description: (1) Notwithstanding anything in this Chapter, the Supreme Court may, in its discretion, grant special leave to appeal from any judgment, decree, determination, sentence or order in any cause or matter passed or made by any court or tribunal in the territory of India. (2) Nothing in clause (1) shall apply to any judgment, determination, sentence or order passed or made by any court or tribunal constituted by or under any law relating to the Armed Forces.
>
> ...
>
> Answer : ['Indian Penal Code, 1860_321', 'Indian Penal Code, 1860_324', 'Indian Penal Code, 1860_302', 'Indian Penal Code, 1860_34']
>
> Explanation (words from the fact statement for each of applicable statute):
>
> Statute: "Indian Penal Code, 1860_34"
> Lines : "the appellants felt offended and  when the members of the complainant party came forward and obstructed the appellant from doing the work and restrained them from pulling out the pipe."
>
> Statute: "Indian Penal Code, 1860_321"
> Lines : " attacked the complainant party.  the first injury to be an incised wound. Second and third were abrasions on the left shoulder and neck. The fourth injury was a lacerated wound on the right parietal area of scalp. On the post-mortem conducted on P4, an incised wound was found on the parietal area of the scalp"
>
> Statute: "Indian Penal Code, 1860_324"
> Lines : " P4 was attacked ...attacked the complainant party. "
>
> #####
>
> Format of Response : Answer containing a list of applicable statutes from the Statute Options above ,and Explanation (containing lines for each of the applicable statutes from the Question) for the Fact Statement delimited by triple backticks (```).
> Note: Response must not include Description from Lecture(description of statute) as Explanation. Do not include statute options in your response.  Lines must be from the Question delimited by triple backticks (```).
>
> Question: ```{Fact_Statement}```
>
> """
> <end of prompt>
>
> Please note that above prompt provides not only the fact text and statute text, but also the extractive explanations from the fact text to train the LLM. For this experiment, 8 cases were used for training the few-shot prompt, while the remaining 37 were used as test cases. *This produced Macro-Precision: 0.41, Macro-Recall:  0.42 and Macro-F1: 0.36, which outperforms the LLM (Template 2) that recorded Macro-Precision: 0.17, Macro-Recall:  0.68 and Macro-F1: 0.27 and InLegalBERT (the best performing baseline) that recorded Macro-Precision: 0.18, Macro-Recall:  0.18 and Macro-F1: 0.17, on the same test set.* However, the quality of explanations generated by this CoT prompt is considerably inferior to those produced by LLM (Template 2). This possibly leads us to the conclusion that few-shot CoT-style prompting, although being trained with explanations, does not necessarily produce high-quality explanations for the test set.
>
>
> E. Judgement Prediction: Why are few-shot examples provided in this setting ? (which is mentioned as training and I assume them to be few-shot examples) Did you try with zero-shot way and they did not perform well ? Any specific reason to consider few-shot in this case and not in statute prediction case?
>
> Response: Yes, we did try zero-shot and noted that it did not perform well. We will add this to the camera-ready version, if accepted.
>
>
>
> F. Line 248-250: Do you mean judgement prediction instead of statute prediction?
>
> Response: Yes. We deeply apologize for the mistake and we will correct it in the camera-ready version, if accepted.
>
>
>
> G. I assume Malik et al. 2021 Dataset of judgement prediction has extractive explanations. Why not prompt LLM for extractive explanations ?
>
> Response: Though the prompt was not directed at producing extractive explanations, it did produce extractive explanations on most of the occasions. Kindly refer to Table 8 in the appendix which shows examples of annotated and generated explanations. E.g. the following is an expert-annotated explanation:
>
> “On a full review of the facts it is clear that this transaction was voluntarily entered into in order indirectly to facilitate the carrying on of the business of the company and was made on the ground of commercial expediency. It therefore falls within the purview of section 12(2) of the Incometax Act, 1922 before its amendment in 1939. This being an investment company, if it borrowed money and utilised the same for its investments on which it earned income, the interest paid by it on the loans will clearly be a permissible deduction under section 12(2) of the Income- tax Act. Whether the loan is taken on an overdraft, or is a fixed deposit or on a debenture makes no difference in law.”
>
> while the corresponding LLM-generated explanation is:
>
> “On a full review of the facts it is clear that this transaction was voluntarily entered into in order indirectly to facilitate the carrying on of the business of the company and was made on the ground of commercial expediency. It therefore falls within the purview of section 12(2) of the Incometax Act, 1922 before its amendment in 1939. This being an investment company, if it borrowed money and utilised the same for its investments on which it earned income, the interest paid by it on the loans will clearly be a permissible deduction under section 12(2) of the Income-tax Act. Whether the loan is taken on an overdraft, or is a fixed deposit or on a debenture makes no difference in law. The only argument urged against allowing this deduction to be made is that the person who took the debentures was the party who sold the ordinary shares. It cannot be disputed that if the debentures were held by a third party, the interest payable on the same would be an allowable deduction in calculating the total income of the assessee company. What difference does it make if the holder of the debentures is a shareholder”
>
> Nevertheless, for the rebuttal, we created another prompt stating in the prompt:
> “Extract the explanation output explicitly from the case proceeding text and not from outside.”
>
> In this process, however, the model's prediction ability was heavily compromised with 60 %  wrong predictions even though the sentences in the explanations matched with the Gold standard ones (not necessarily better than the earlier results).
>
>
> Additional comments:
>
> About the small datasize:
>
> We acknowledge the concern of the reviewer. As already mentioned in the limitations, due to the token limitation and high subscription charges of paid LLMs, we could not do experiments on bigger test datasets. Also, legal expert annotations are expensive and time-consuming to obtain. However, to our knowledge, we have used some of the biggest legal expert annotated datasets available for Indian cases, in this paper. We also will release an expert annotated dataset with this paper, upon acceptance.
>
> About data contamination issue if the LLM's have already seen the cases:
>
> Kindly note that the training set of cases (for few-shot) is always disjoint from the test set of cases. So, there should not be any data contamination.
>
> About analysis on generated explanations:
>
> In the rebuttal we have provided some analysis of the generated explanations esp. w.r.t. the extractive nature of the same. Kindly refer to Tables 7 and 8 in the appendix where we have shown some examples of generated explanations along with the annotated explanations. We will try to add more insights on the explanations in the camera-ready version, if accepted.
>
> About study of hallucinations:
>
> We request the reviewer to kindly refer to Tables 11 and 12 in the appendix where we have tried to capture the prominent examples of hallucinations observed for the statute prediction and explanation task. The hallucinations mostly comprise: generation of wrong statutes based on superficial features like words found in the fact text and not always judging the applicability of the same for the case. We could not provide a more detailed analysis of hallucinations, which we identified in the course of our experiments, in the main paper due to the lack of space.
>
> About Typos Grammar Style And Presentation Improvements: We thank the reviewer for pointing these out. We will correct this in the camera-ready version, if accepted.
>
>
> Additional experiments:
>
> For judgement prediction task, we reversed the order of the prediction label and the explanation in the training prompt (that is first explanation and then prediction label) in the few-shot training and requested the corresponding output from the LLM. We have drawn inspiration from the paper Chain-of-Thought Prompting Elicits Reasoning in Large Language Models. Wei et al. 2022, Section 3.3 Ablation Study, Chain of thought after answer, in order to analyse the sequential reasoning embodied in the explanation and its dependence/consequence on the prediction ability of the LLM model beyond just extracting knowledge from the input text.
> The output explanations were obtained for 80% and were logically reasoned. But, still, only 20% was predicted correctly by the model. This therefore led to a deterioration in the performance when compared with our previous prompt style where prediction *preceded* explanation.
>
> Therefore, from these sets of experimentations, we can safely say that the prompting style used by us previously in the paper was already in alignment with the state-of-the art style (like CoT).

---

### Official Review · Reviewer_ZNGP · 2023-08-03

**Typos Grammar Style And Presentation Improvements:** 1. Some spacing issues, e.g., spacing…
**Soundness:** 3

**Excitement:**

3: Ambivalent: It has merits (e.g., it reports state-of-the-art results, the idea is nice), but there are key weaknesses (e.g., it describes incremental work), and it can significantly benefit from another round of revision. However, I won't object to accepting it if my co-reviewers champion it.

**Paper Topic And Main Contributions:**

This paper examines the performance of recent large language models, specifically GPT 3.5 series from OpenAI, on the legal domain tasks. They evaluate the GPT 3.5 for two tasks, statute prediction and judegement prediction. For the multi-label classification task of statute prediction, they evaluated text-davinci-003 on 45 texts from FIRE AILA 2019 (Bhattacharya et al., 2019a) and 200 random cases from Indian Supreme Court. For the binary classification task of judegement prediction (allowed or dismissed), they evaluate GPT 3.5 turbo on 256 cases from Malik et al. (2021). They also manually annotated a small number of cases to find the explanations in the context. The authors compare the performance of GPT 3.5 with several fine-tuned language models such as BERT for the classification tasks. The experimental results show that GPT 3.5 performs better than fine-tuned models on statute prediction, but worse on judgement prediction. The authors also discuss the evidence of gender and region bias in the GPT predictions and generated explanations, and further convey ethical concerns of deploying LLMs in the legal domain.

**Questions For The Authors:**

Question A:

It is interesting to see that, in Table 3, LLM's performance on the judegement prediction without explanation is considerablely lower than prediction with explanation. Did you try prediction with explanation for all 256 cases? would it have the same trend (better performance than prediction only (256 cases))?

Question B:

Have you tried SOTA extractive QA systems for the explanation task since the task seems comparable?

**Reasons To Accept:**

1. The paper evaluates the recent GPT 3.5 models on the legal domain tasks.

2. The annotated explanations, though small, could be helpful for other researchers who are interested in Indian legal cases in the community.

3. The analysis of generated explanations by GPT 3.5 compared to annotated explanations by legal experts, provides insights into the potential bias in the LLMs generations.

**Reasons To Reject:**

1. The paper is not easy to follow. For example, the introduction of the dataset is not self-contained. And the analysis of bias also deserves more discussion.

2. The paper lacks clarity in the technical part as well, e.g., how do you select the hyper-parameters since there are no development set? Line 100, "We observed that the best results on our dataset are provided by text-davinci-003 (paid version) and so all results reported in this paper are produced by this model." How does text-davinci-003 compare to other models? Does it significantly outperform others or do they have comparable results?

3. The authors do not include a good brief of related work in the legal domain, or a comparison with previous SOTA models.

**Reproducibility:**

3: Could reproduce the results with some difficulty. The settings of parameters are underspecified or subjectively determined; the training/evaluation data are not widely available.

**Reviewer Confidence:**

4: Quite sure. I tried to check the important points carefully. It's unlikely, though conceivable, that I missed something that should affect my ratings.

---

> ### Author Rebuttal · Authors · 2023-08-29
>
> We thank the reviewer for the valuable comments. We have tried our best to address them and hope that the reviewer will now have a better understanding of the paper.
>
> Question A:
> It is interesting to see that, in Table 3, LLM's performance on the judgment prediction without explanation is considerablely lower than prediction with explanation. Did you try prediction with explanation for all 256 cases? would it have the same trend (better performance than prediction only (256 cases))?
>
>
> Response: No, we tried it in only 56 cases as we have expert annotations for only 56 cases. In the future, we look forward to expanding the annotated set and performing the experiments as suggested by the reviewer.
>
>
> Question B:
> Have you tried SOTA extractive QA systems for the explanation task since the task seems comparable?
>
> Response:
>
> To our knowledge, the prompts used in this paper are novel as they have not been used before on the two tasks presented in the paper. Yu et al., 2022 (cited in the paper) used zero-shot Chain-Of-Thought (CoT) prompting for statute prediction, where they first generate explanation and then used it for prediction; this did not produce the best results (than without explanations) and also they did not present any analysis of the quality of explanations. We, however, have integrated prediction and explanation as a part of the prompts (in both the templates 1 and 2 by stating additionally “extract words or lines from the case due to which the predicted statutes are applicable”; kindly see Tables 5 and 6 in the appendix).
>
> For statute prediction, to adhere to the more conventional few-shot CoT prompting (Chain-of-Thought Prompting Elicits Reasoning in Large Language Models. Wei et al. 2022) where the predictions with explanations are a part of few-shot prompting training, for the rebuttal, we have designed a few-shot prompt for statute prediction as below:
>
> <start of prompt>
>
> prompt = f"""Given a fact statement as a Question delimited by triple backticks (```) and multiple choice options representing the names of statutes, predict the applicable statutes from the Statute Options, and the explanation for each of the applicable statute containing lines from the Question delimited by triple backticks(```). Example is given for your reference. The lecture provides a description of each statute, and your task is to select the statutes from Statute Options that best align with the words from the given Question. Choose the most relevant statutes and lines that match the given context. Statute options (Answer must always be from these) and Lecture (description of statute) are same for each Question. Answer must always be from Statute Options.
>
> Example :
>
> Fact statement: "These appeals are directed against the judgment of a High Court whereby an appeal and a criminal revision were disposed of. The appellants were found guilty and sentenced to undergo various terms of sentences. The Criminal Appeal was filed by three appellants questioning the conviction and sentence as recorded. Complainant filed a revision petition stating that she was entitled to compensation. Background facts giving rise to the trial are essentially as follows: .."
>
> Statute options (Answer must always be from these):
>
> 1. "Constitution_226"
> 2. "Constitution_136"
> ...
>
>
> Lecture (description of statute):
>
> 1."Constitution_226":
> Title: Power of High Courts to issue certain writs
> Description: (1) Notwithstanding anything in Article 32 every High Court shall have powers, throughout the territories in relation to which it exercise jurisdiction, to issue to any person or authority, including in appropriate cases, any Government, within those territories directions, orders or writs, including writs in the nature of habeas corpus, mandamus, prohibitions, quo warranto and certiorari, or any of them, for the enforcement of any of the rights conferred by Part III and for any other purpose
>
> 2. "Constitution_136":
> Title: Special leave to appeal by the Supreme Court
> Description: (1) Notwithstanding anything in this Chapter, the Supreme Court may, in its discretion, grant special leave to appeal from any judgment, decree, determination, sentence or order in any cause or matter passed or made by any court or tribunal in the territory of India. (2) Nothing in clause (1) shall apply to any judgment, determination, sentence or order passed or made by any court or tribunal constituted by or under any law relating to the Armed Forces.
>
> ...
>
> Answer : ['Indian Penal Code, 1860_321', 'Indian Penal Code, 1860_324', 'Indian Penal Code, 1860_302', 'Indian Penal Code, 1860_34']
>
> Explanation (words from the fact statement for each of applicable statute):
>
> Statute: "Indian Penal Code, 1860_34"
> Lines : "the appellants felt offended and  when the members of the complainant party came forward and obstructed the appellant from doing the work and restrained them from pulling out the pipe."
>
> Statute: "Indian Penal Code, 1860_321"
> Lines : " attacked the complainant party.  the first injury to be an incised wound. Second and third were abrasions on the left shoulder and neck. The fourth injury was a lacerated wound on the right parietal area of scalp. On the post-mortem conducted on P4, an incised wound was found on the parietal area of the scalp"
>
> Statute: "Indian Penal Code, 1860_324"
> Lines : " P4 was attacked ...attacked the complainant party. "
>
> #####
>
> Format of Response : Answer containing a list of applicable statutes from the Statute Options above ,and Explanation (containing lines for each of the applicable statutes from the Question) for the Fact Statement delimited by triple backticks (```).
> Note: Response must not include Description from Lecture(description of statute) as Explanation. Do not include statute options in your response.  Lines must be from the Question delimited by triple backticks (```).
>
> Question: ```{Fact_Statement}```
>
> """
> <end of prompt>
>
> Please note that above prompt provides not only the fact text and statute text, but also the extractive explanations from the fact text to train the LLM. For this experiment, 8 cases were used for training the few-shot prompt, while the remaining 37 were used as test cases. *This produced Macro-Precision: 0.41, Macro-Recall:  0.42 and Macro-F1: 0.36, which outperforms the LLM (Template 2) that recorded Macro-Precision: 0.17, Macro-Recall:  0.68 and Macro-F1: 0.27 and InLegalBERT (the best performing baseline) that recorded Macro-Precision: 0.18, Macro-Recall:  0.18 and Macro-F1: 0.17, on the same test set*. However, the quality of explanations generated by this CoT prompt is considerably inferior to those produced by LLM (Template 2). This possibly leads us to the conclusion that few shot CoT-style prompting, although being trained with explanations, does not necessarily produce high-quality explanations for the test set.
>
>
> Kindly note that, for judgement prediction and explanation have already adhered to the more conventional few-shot CoT prompting (Chain-of-Thought Prompting Elicits Reasoning in Large Language Models. Wei et al. 2022) where the predictions with explanations are a part of few-shot prompting training. Kindly refer to Table 9 in the appendix to note that for a given case, the explanation along with the actual prediction is provided in the few-shot prompt training. To our knowledge, few-shot CoT prompting has been under-explored for legal tasks. On the whole, we feel, our proposed prompts are novel with regard to the tasks presented in the paper.
>
> Additional comments:
>
> Regarding the paper being not easy to follow: We will try our best to improve the paper in the camera-ready version, if accepted. We are apologetic for keeping the bias and fairness section (Section 4) short due to the paucity of space in the main paper. However, we have tried to connect the discussion in the main paper with the supporting results in the appendix (kindly refer to Sections D and E in the appendix). To our knowledge, this is among the first works that discuss bias, fairness and ethics w.r.t. statute prediction and judgement prediction esp. on Indian cases.
>
> Regarding the paper lacking technical clarity: We have mostly kept the hyper-parameters to their default values. Since we used subscribed version of GPT, we could not perform extensive experiments with all the models like text-davinci-003 due to high cost. We did some initial experiments with randomly chosen sets of 5 cases and tried them on various models and chose the one that produced the best results. The paid GPT models handsomely outperformed the free models like BLOOM, FLAN-T5 and LLaMA. Among paid GPT models, we chose the numerically better performing models though the performances were not necessarily significantly different.
>
> About the related work being insufficient and lack of comparison with SOTA (LLM?) models: Due to the paucity of space in the main paper, we have restricted to citing only the directly relevant works. We will try to add a more detailed literature study in the camera-ready version, if accepted. In this rebuttal, we have added a qualitative comparison of a close prior work.
>
> About Typos Grammar Style And Presentation Improvements: We thank the reviewer for pointing these out. We will correct this in the camera-ready version, if accepted.
>
> Additional experiments:
>
> For judgement prediction task, we reversed the order of the prediction label and the explanation in the training prompt (that is first explanation and then prediction label) in the few-shot training and requested the corresponding output from the LLM. We have drawn inspiration from from the paper Chain-of-Thought Prompting Elicits Reasoning in Large Language Models. Wei et al. 2022, Section 3.3 Ablation Study, Chain of thought after answer, in order to analyse the sequential reasoning embodied in the explanation and its dependence/consequence on the prediction ability of the LLM model beyond just extracting knowledge from the input text.
> The output explanations were obtained for 80% and were logically reasoned. But, still, only 20% was predicted correctly by the model. This therefore led to a deterioration in the performance when compared with our previous prompt style where prediction *preceded* explanation.
>
> Therefore, from these sets of experimentations, we can safely say that the prompting style used by us previously in the paper was already in alignment with the state-of-the art style (like CoT).

---

### Official Review · Reviewer_Je4i · 2023-08-14

**Soundness:** 2

**Excitement:**

2: Mediocre: This paper makes marginal contributions (vs non-contemporaneous work), so I would rather not see it in the conference.

**Missing References:**




**Paper Topic And Main Contributions:**

This paper evaluates LLMs and supervised baselines on two India legal tasks: statute prediction and judgment prediction.

**Questions For The Authors:**

1. For LLM prompting, do you use zero-shot prompting or few-shot prompting? What type of statute information is include into the prompts, e.g., statute number, name or content? Do you use similar cases for prediction statute?
2. What is the meaning of the “Prediction with explanation” setting for supervised baselines? These baselines, e.g., XLNet, cannot generate explanations.
3. Why judgment prediction is a binary classification task? It differs from other literature, e.g., "Legal Judgment Prediction via Topological Learning" and "Neural Legal Judgment Prediction in English"

**Reasons To Accept:**

1. This work focuses on the legal tasks of India, which is not extensively studied.
2. This work analyzes the quality of LLM generated explanations and compares them with expert annotations.


**Reasons To Reject:**

1. There is not novelty in methodology.
2. The size of test dataset is small.
3. The paper claims to evaluate several LLMs, but only report the performance of one model for each task.
4. The results are doubtful. For statute prediction, the paper indicates that davinci is better than GPT-3.5 turbo (Line 102), while for judgment prediction, GPT-3.5 is better (Line 220-222). Besides, the baseline performances in Table 1 are extremely low (macro prec around 0.1), while these baselines are claimed to be trained on 18021 samples. There could be huge distribution shifts between the training and test data. However, this is not discussed at all.
5. The prompt design is not well illustrated.


**Reproducibility:**

2: Would be hard pressed to reproduce the results. The contribution depends on data that are simply not available outside the author's institution or consortium; not enough details are provided.

**Reviewer Confidence:**

4: Quite sure. I tried to check the important points carefully. It's unlikely, though conceivable, that I missed something that should affect my ratings.

**Typos Grammar Style And Presentation Improvements:**

Line 28: wrong citation format. Remove the parentheses.
Line 123-124: “is was” -> “is”

---

> ### Author Rebuttal · Authors · 2023-08-29
>
> We thank the reviewer for the valuable comments. We have tried our best to address them and hope the reviewer will now have a better understanding of the work.
>
>
> Q. For LLM prompting, do you use zero-shot prompting or few-shot prompting?
>
> Response: We have tried both zero-shot (statute prediction) and few-shot (for judgment prediction). The prompts used in this paper are shown in Tables 5, 6, 9 and 10 in the appendix. For the rebuttal, we have also tried few-shot Chain-of-Thought (CoT) prompting    (described later).
>
>
> Q. What type of statute information is include into the prompts, e.g., statute number, name or content?
>
> Response: Statute ids (unique identifier), title (brief overview of the statute) and description (more detailed description of the statute) are provided in the prompt. Kindly refer to Tables 5 and 6 in the appendix.
>
> Q. Do you use similar cases for prediction statute?
>
> Response: We have tried training cases, which may not be similar to a current test case, on one of the prompts (Template 1, which did NOT produce better results than Template 2, whose performance is reported in the paper). Kindly refer to Tables 5 and 6 in the appendix.
>
>
> Q. What is the meaning of the “Prediction with explanation” setting for supervised baselines? These baselines, e.g., XLNet, cannot generate explanations.
>
> Response:
>
> We agree with the reviewer. However, we draw attention to the fact that generating explanations along with predictions is a challenging task and to our knowledge, this has not been tried in Law-AI applications. The setup will possibly be similar to that of Task 2 as reported by Lamm et al. in QED: A Framework and Dataset for Explanations in Question Answering (https://arxiv.org/abs/2009.06354). However, a major challenge in the legal domain is building large enough datasets (with explanations manually annotated) for such complex tasks. We look forward to implementing such a model in the future after expanding the current dataset. Nevertheless, with our explanation annotations, we have been able to apply CoT-style prompting on our dataset (described later) to generate explanations with the prediction.
>
>
> Q. Why judgment prediction is a binary classification task? It differs from other literature, e.g., "Legal Judgment Prediction via Topological Learning" and "Neural Legal Judgment Prediction in English"
>
> Response: We have used the CJPE setup by Malik et al. 2021 where they have treated judgement prediction as a binary classification task.
>
> Additional comments:
>
> 1. About the lack of novelty:
>
> To our knowledge, the prompts used in this paper are novel as they have not been used before on the two tasks presented in the paper (particularly for Indian datasets). Yu et al., 2022 (cited in the paper) used zero-shot Chain-Of-Thought (CoT) prompting for statute prediction, where they first generated explanation and then used it for prediction; this did not produce the best results (than without explanations) and also they did not present any analysis of the quality of explanations. We, however, have integrated prediction and explanation as a part of the prompts (in both the templates 1 and 2 by stating additionally “extract words or lines from the case due to which the predicted statutes are applicable”; kindly see Tables 5 and 6 in the appendix).
>
> For statute prediction, to adhere to the more conventional few-shot CoT prompting (Chain-of-Thought Prompting Elicits Reasoning in Large Language Models. Wei et al. 2022) where the predictions with explanations are a part of few-shot prompting training, for the rebuttal, we have designed a few-shot prompt for statute prediction as below:
>
> <start of prompt>
>
> prompt = f"""Given a fact statement as a Question delimited by triple backticks (```) and multiple choice options representing the names of statutes, predict the applicable statutes from the Statute Options, and the explanation for each of the applicable statute containing lines from the Question delimited by triple backticks(```). Example is given for your reference. The lecture provides a description of each statute, and your task is to select the statutes from Statute Options that best align with the words from the given Question. Choose the most relevant statutes and lines that match the given context. Statute options (Answer must always be from these) and Lecture (description of statute) are same for each Question. Answer must always be from Statute Options.
>
> Example :
>
> Fact statement: "These appeals are directed against the judgment of a High Court whereby an appeal and a criminal revision were disposed of. The appellants were found guilty and sentenced to undergo various terms of sentences. The Criminal Appeal was filed by three appellants questioning the conviction and sentence as recorded. Complainant filed a revision petition stating that she was entitled to compensation. Background facts giving rise to the trial are essentially as follows: .."
>
> Statute options (Answer must always be from these):
>
> 1. "Constitution_226"
> 2. "Constitution_136"
> ...
>
>
> Lecture (description of statute):
>
> 1."Constitution_226":
> Title: Power of High Courts to issue certain writs
> Description: (1) Notwithstanding anything in Article 32 every High Court shall have powers, throughout the territories in relation to which it exercise jurisdiction, to issue to any person or authority, including in appropriate cases, any Government, within those territories directions, orders or writs, including writs in the nature of habeas corpus, mandamus, prohibitions, quo warranto and certiorari, or any of them, for the enforcement of any of the rights conferred by Part III and for any other purpose
>
> 2. "Constitution_136":
> Title: Special leave to appeal by the Supreme Court
> Description: (1) Notwithstanding anything in this Chapter, the Supreme Court may, in its discretion, grant special leave to appeal from any judgment, decree, determination, sentence or order in any cause or matter passed or made by any court or tribunal in the territory of India. (2) Nothing in clause (1) shall apply to any judgment, determination, sentence or order passed or made by any court or tribunal constituted by or under any law relating to the Armed Forces.
>
> ...
>
> Answer : ['Indian Penal Code, 1860_321', 'Indian Penal Code, 1860_324', 'Indian Penal Code, 1860_302', 'Indian Penal Code, 1860_34']
>
> Explanation (words from the fact statement for each of applicable statute):
>
> Statute: "Indian Penal Code, 1860_34"
> Lines : "the appellants felt offended and  when the members of the complainant party came forward and obstructed the appellant from doing the work and restrained them from pulling out the pipe."
>
> Statute: "Indian Penal Code, 1860_321"
> Lines : " attacked the complainant party.  the first injury to be an incised wound. Second and third were abrasions on the left shoulder and neck. The fourth injury was a lacerated wound on the right parietal area of scalp. On the post-mortem conducted on P4, an incised wound was found on the parietal area of the scalp"
>
> Statute: "Indian Penal Code, 1860_324"
> Lines : " P4 was attacked ...attacked the complainant party. "
>
> #####
>
> Format of Response : Answer containing a list of applicable statutes from the Statute Options above ,and Explanation (containing lines for each of the applicable statutes from the Question) for the Fact Statement delimited by triple backticks (```).
> Note: Response must not include Description from Lecture(description of statute) as Explanation. Do not include statute options in your response.  Lines must be from the Question delimited by triple backticks (```).
>
> Question: ```{Fact_Statement}```
>
> """
> <end of prompt>
>
> Please note that above prompt provides not only the fact text and statute text, but also the extractive explanations from the fact text to train the LLM. For this experiment, 8 cases were used for training the few-shot prompt, while the remaining 37 were used as test cases. *This produced Macro-Precision: 0.41, Macro-Recall:  0.42 and Macro-F1: 0.36, which outperforms the LLM (Template 2) that recorded Macro-Precision: 0.17, Macro-Recall:  0.68 and Macro-F1: 0.27 and InLegalBERT (the best performing baseline) that recorded Macro-Precision: 0.18, Macro-Recall:  0.18 and Macro-F1: 0.17, on the same test set.* However, the quality of explanations generated by this CoT prompt is considerably inferior to those produced by LLM (Template 2). This possibly leads us to the conclusion that few-shot CoT-style prompting, although being trained with explanations, does not necessarily produce high-quality explanations for the test set.
>
> Kindly note that, for judgement prediction and explanation we have already adhered to the more conventional few-shot CoT prompting (Chain-of-Thought Prompting Elicits Reasoning in Large Language Models. Wei et al. 2022) where the predictions with explanations are a part of few-shot prompting training. Kindly refer to Table 9 in the appendix to note that for a given case, the explanation along with the actual prediction is provided in the few-shot prompt training. To our knowledge, few-shot CoT prompting has been under-explored for legal tasks. On the whole, we feel, our proposed prompts are novel with regard to the tasks presented in the paper.
>
>
> 2. About the size of test dataset being small:
>
> We acknowledge the concern of the reviewer. As already mentioned in the limitations, due to the token limitation and high subscription charges of paid LLMs (the better results were produced by the paid models), we could not do experiments on bigger test datasets. Also, legal expert annotations are expensive and time-consuming to obtain. However, to our knowledge, we have used some of the biggest legal expert annotated datasets available for Indian cases, in this paper. We also will release an expert annotated dataset with this paper, upon acceptance.
>
>
> 3. About the paper reporting the performance of one LLM model for each task:
>
> We acknowledge the concern of the reviewer. Since we used subscribed version of GPT, we could not perform extensive experiments with all the models like text-davinci-003 due to high cost. We did some initial experiments with randomly chosen sets of 5 cases and tried them on various models and chose the one that produced the best results. The paid GPT models handsomely outperformed the free models like BLOOM, FLAN-T5 and LLaMA. Among paid GPT models, we chose the numerically better performing model though the performances were not necessarily significantly different.
>
> 4.  About the results being doubtful:
>
> For statute prediction, the pretrained models (InLegalBERT, LEGAL-BERT and XLNet) were actually fine-tuned (and not trained) on 18021 cases (lines 111-115 in the paper). Kindly also note that InLegalBERT is pre-trained on Indian legal documents and hence is expected to have the necessary context to act as a strong baseline for both our tasks.
>
> 5.  About the  prompt design being not well illustrated:
>
> We could not extensively explain the prompts in the main paper due to the lack of space. However, we have tried our best to motivate our prompts in the main paper and provided the exact prompts used in the appendix (kindly see Tables 5, 6, 9 and 10 in the appendix). We have also used a new prompt on few-shot Chain-Of-Thought in the rebuttal with the performance (already discussed), which we will add to the camera-ready version, if accepted.
>
> 6. About Typos Grammar Style And Presentation Improvements: We thank the reviewer for pointing these out. We will correct this in the camera-ready version, if accepted.
>
> Additional experiments:
>
> For judgement prediction task, we reversed the order of the prediction label and the explanation in the training prompt (that is first explanation and then prediction label) in the few-shot training and requested the corresponding output from the LLM. We have drawn inspiration from the paper Chain-of-Thought Prompting Elicits Reasoning in Large Language Models. Wei et al. 2022, Section 3.3 Ablation Study, Chain of thought after answer, in order to analyse the sequential reasoning embodied in the explanation and its dependence/consequence on the prediction ability of the LLM model beyond just extracting knowledge from the input text.
> The output explanations were obtained for 80% and were logically reasoned. But, still, only 20% was predicted correctly by the model. This therefore led to a deterioration in the performance when compared with our previous prompt style where prediction *preceded* explanation.
>
> Therefore, from these sets of experimentations, we can safely say that the prompting style used by us previously in the paper was already in alignment with the state-of-the art style (like CoT).

---

### Meta-Review · Area_Chair_m8QJ · 2023-09-19

**Recommendation:** 3

**Metareview:**

This paper evaluates the performance of large language models (LLMs) on two legal tasks: statute prediction and judgment prediction. The reviewers appreciate the focus on the Indian legal system, which is not extensively studied, and the comparison of LLM-generated explanations with expert annotations. However, they raise concerns about the lack of novelty in methodology, the small size of the test dataset, and the clarity of the paper. The authors' rebuttal addresses these concerns, providing detailed explanations and additional experiments. The reviewers also suggest improvements in the presentation and request further clarification on certain aspects of the study. Overall, the paper is seen as a valuable contribution to the field, despite some weaknesses.

---

### Decision · Program_Chairs · 2023-10-07

**Decision:**

Accept-Findings

**Comment:**

This paper evaluates the performance of large language models (LLMs) on two legal tasks: statute prediction and judgment prediction. The reviewers appreciate the focus on the Indian legal system, which is not extensively studied, and the comparison of LLM-generated explanations with expert annotations. However, they raise concerns about the lack of novelty in methodology, the small size of the test dataset, and the clarity of the paper. The authors' rebuttal addresses these concerns, providing detailed explanations and additional experiments. The reviewers also suggest improvements in the presentation and request further clarification on certain aspects of the study. Overall, the paper is seen as a valuable contribution to the field, despite some weaknesses.